# TENSORIAL MIXTURE MODELS

**Or Sharir, Ronen Tamari, Nadav Cohen & Amnon Shashua**
The Hebrew University of Jerusalem
`{or.sharir,ronent,cohennadav,shashua}@cs.huji.ac.il`

## ABSTRACT

We introduce a generative model, we call Tensorial Mixture Models (TMMs) based on mixtures of basic component distributions over local structures (e.g. patches in an image) where the dependencies between the local-structures are represented by a "priors tensor" holding the prior probabilities of assigning a component distribution to each local-structure.

In their general form, TMMs are intractable as the priors tensor is typically of exponential size. However, when the priors tensor is decomposed it gives rise to an arithmetic circuit which in turn transforms the TMM into a Convolutional Arithmetic Circuit (ConvAC). A ConvAC corresponds to a shallow (single hidden layer) network when the priors tensor is decomposed by a CP (sum of rank-1) approach and corresponds to a deep network when the decomposition follows the Hierarchical Tucker (HT) model.

The ConvAC representation of a TMM possesses several attractive properties. First, the inference is tractable and is implemented by a forward pass through a deep network. Second, the architectural design of the model follows the deep networks community design, i.e., the structure of TMMs is determined by just two easily understood factors: size of pooling windows and number of channels. Finally, we demonstrate the effectiveness of our model when tackling the problem of classification with missing data, leveraging TMMs unique ability of tractable marginalization which leads to optimal classifiers regardless of the missingness distribution.

## 1 INTRODUCTION

Generative models have played a crucial part in the early development of the field of Machine Learning. However, in recent years they were mostly cast aside in favor of discriminative models, lead by the rise of ConvNets (LeCun et al., 2015), which were found to perform equally well or better than classical generative counter-parts on almost any task. Despite the increased interest in unsupervised learning, many of the recent studies on generative models choose to focus solely on the generation capabilities of these models (Goodfellow et al., 2014; Gregor et al., 2015; van den Oord et al., 2016; Dinh et al., 2016; Tran et al., 2016; Chen et al., 2016; Kingma et al., 2016; Kim and Bengio, 2016). There is much less emphasis on leveraging generative models to solve actual tasks, e.g. semi-supervised learning (Kingma et al., 2014; Springenberg, 2016; Maaløe et al., 2016; Forster et al., 2015; Salimans et al., 2016), image restoration (Dinh et al., 2014; Bengio et al., 2014; van den Oord et al., 2016; Zoran and Weiss, 2011; Rosenbaum and Weiss, 2015; Sohl-Dickstein et al., 2015; Theis and Bethge, 2015) or unsupervised feature representation (Radford et al., 2016; Coates et al., 2011). Nevertheless, work on generative models for solving actual problems are yet to show a meaningful advantage over competing discriminative models.

On the most fundamental level, the difference between a generative model and a discriminative one is simply the difference between learning $P(X, Y)$ and learning $P(Y|X)$, respectively. While it is always possible to infer $P(Y|X)$ given $P(X, Y)$, it might not be immediately apparent why the generative objective is preferred over the discriminative one. In Ng and Jordan (2002), this question was studied w.r.t. the sample complexity, proving that under some cases it can be significantly lesser in favor of the generative classifier. However, their analysis was limited only to specific pairs of discriminative and generative classifiers, and they did not present a general case where the the generative method is undeniably preferred. We wish to highlight one such case, where learning

$P(X, Y)$ is provenly better regardless of the models in question, by examining the problem of classification with missing data. Despite the artificially well-behave nature of the typical classification benchmarks presented in current publications, real-world data is usually riddled with noise and missing values – instead of observing $X$ we only have a partial observation $\hat{X}$ – a situation that tends to be ignored in modern research. Discriminative models have no natural mechanisms to handle missing data and instead must rely on data imputation, i.e. filling missing data by a preprocessing step prior to prediction. Unlike the discriminative approaches, generative models are naturally fitted to handle missing data by simply marginalizing over the unknown values in $P(X, Y)$, from which we can attain $P(Y|\hat{X})$ by an application of Bayes Rule. Moreover, under mild assumptions which apply to many real-world settings, this method is proven to be optimal *regardless of the process by which values become missing* (see sec. 5 for a more detailed discussion).

While almost all generative models can represent $P(X, Y)$, only few can actually infer its exact value efficiently. Models which possess this property are said to have *tractable inference*. Many studies specifically address the hard problem of learning generative models that do not have this property. Notable amongst those are works based on Variational Inference (Kingma and Welling, 2014; Kingma et al., 2014; Blei et al., 2003; Wang and Grimson, 2007; Makhzani et al., 2015; Kingma et al., 2016), which only provide approximated inference, and ones based on Generative Adversarial Networks (Goodfellow et al., 2014; Radford et al., 2016; Springenberg, 2016; Chen et al., 2016; Salimans et al., 2016; Makhzani et al., 2015), which completely circumvent the inference problem by restructuring the learning problem as a two-player game of discriminative objectives – both of these approaches are incapable of tractable inference.

There are several advantages to models with tractable inference (e.g. they could be simpler to train), and as we have shown above, this property is also a requirement for proper handling of missing data in the form of marginalization. In practice, to marginalize over $P(X, Y)$ means to perform integration on it, thus, even if it is tractable to compute $P(X, Y)$, it still might not be tractable to compute every possible marginalization. Models which are capable of this are said to have *tractable marginalization*. Mixture Models (e.g. Gaussian Mixture Models) are the classical example of a generative model with tractable inference, as well as tractable marginalization. Though they are simple to understand, easy to train and even known to be universal – can approximate any distribution given sufficient capacity – they do not scale well to high-dimensional data. The Gaussian Mixture Model is an example of a shallow model – containing just a single latent variable – with limited expressive efficiency. More generally, Graphical Models are deep and exponentially more expressive, capable of representing intricate relations between many latent variables. While not all kinds of Graphical Models are tractable, many are, e.g. Latent Tree Models (Zhang, 2004; Mourad et al., 2013) and Sum-Product Networks (Poon and Domingos, 2011). The main issue with generic graphical models is that by virtue of being too general they lack the inductive bias needed to efficiently model unstructured data, e.g. images or text. Despite the success of structure learning algorithms (Huang et al., 2015; Gens and Domingos, 2013; Adel et al., 2015) on structured datasets, such as discovering a hierarchy among diseases in patients health records, there are no similar results on unstructured datasets. Indeed some recent works on the subject have failed to solve even simple handwritten digit classification tasks (Adel et al., 2015). Thus deploying graphical models on such cases requires experts to manually design the model. Other attempts which harness neural networks blocks (Dinh et al., 2014; 2016) offer tractable inference, but not tractable marginalization.

To summarize, most generative models do not have tractable inference, and of the few models which do, they all possess one or more of the following shortcomings: (i) they do not possess the expressive capacity to model high-dimensional data (e.g. images), (ii) they require explicitly designing all the dependencies of the data, or (iii) they do not have tractable marginalization.

We present in this paper a family of generative models we call *Tensorial Mixture Models* (TMMs), which aim to address the above shortcomings of alternative models. Under TMMs, we assume that the data generated by our model is composed of a sequence of local-structures (e.g. patches in an image), where each local-structure is generated from a small set of simple component distributions (e.g. Gaussian), and the dependencies between the local-structures are represented by a *prior tensor* holding the prior probabilities of assigning a component distribution to each local-structure. In their general form, TMMs are intractable as the prior tensor is typically of exponential size. However, by *decomposing the prior tensor*, inference of TMMs becomes realizable by *Convolutional Arithmetic Circuits* (ConvACs) – a recently proposed (Cohen et al., 2016a) ConvNet architecture based on two

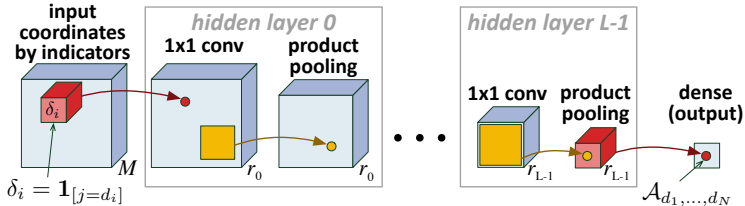

Figure 1: The decoding algorithm of an arbitrary tensor decomposition represented by a ConvAC.

operations, weighted sum and product pooling – which enables both tractable inference as well as tractable marginalization. While Graphical Models are typically hard to design, ConvACs follow the same design conventions of modern ConvNets, which reduces the task of designing a model to simply choosing the number of channels at each layer, and size of pooling windows. ConvACs were also the subject of several theoretical studies on its expressive capacity (Cohen et al., 2016a; Cohen and Shashua, 2016b) and comparing them to ConvNets (Cohen and Shashua, 2016a), showing they are especially suitable for high-dimensional natural data (images, audio, etc.) with a non-negligible advantage over standard ConvNets. Sum-Product Networks are another kind of Graphical Model realizable by Arithmetic Circuits, but they do not posses the same theoretical guarantees, nor do they provide a simple method to design efficient and expressive models.

The rest of the article is organized as follows. In sec. 2 we briefly review mathematical background on tensors required in order to follow our work. This is followed by sec. 3 which presents our generative model and its theoretical properties. How our model is trained is covered in sec. 4, and a thorough discussion on the importance of marginalization and its implications on our model is given in sec. 5. We conclude the article by presenting our experiments on classification with missing data in sec. 6, and revisit the main points of the article and future research in sec. 7.

## 2 PRELIMINARIES

We begin by establishing the minimal background in the field of tensor analysis required for following our work (see app. A for a more detailed review of the subject). A tensor is best thought of as a multi-dimensional array $\mathcal{A}_{d_1,\ldots,d_N} \in \mathbb{R}$, where $\forall i \in [N], d_i \in [M_i]$ and $N$ is referred to as the *order* of the tensor. For our purposes we typically assume that $M_1 = \ldots = M_N = M$, and denote it as $\mathcal{A} \in (\mathbb{R}^M)^{\otimes N}$. It is immediately apparent that performing operations with tensors, or simply storing them, quickly becomes intractable due to their exponential size of $M^N$. That is one of the primary motivations behind tensor decomposition, which can be seen as a generalization of low-rank matrix factorization.

The relationship between tensor decomposition and networks arises from the simple observation, that through decomposition one can tradeoff storage complexity with computation, where the type of computation consists of sums and products. Specifically, the decompositions could be described by a compact representation coupled with a decoding algorithm of polynomial complexity to retrieve the entries of the tensor. Most tensor decompositions have a decoding algorithm representable via computation graphs of products and weighted sums, also known as *Arithmetic Circuits* (Shpilka and Yehudayoff, 2010) or Sum-Product Networks (Poon and Domingos, 2011). More specifically, these circuits take as input $N$ indicator vectors $\delta_1, \ldots, \delta_N$, representing the coordinates $(d_1, \ldots, d_N)$, where $\delta_i = \mathbf{1}_{[j=d_i]}$, and output the value of $\mathcal{A}_{d_1,\ldots,d_N}$, where the weights of these circuits form the compact representation of tensors.

Applying this perspective to two of the most common decomposition formats, CANDE-COMP/PARFAC (CP) and Hierarchical Tucker (HT), give rise to a shared framework for representing their decoding circuits by convolutional networks as illustrated in fig. 1, where a shallow network with one hidden layer corresponds to the CP decomposition, and a deep network with $\log_2(N)$ hidden layers corresponds to the HT decomposition. The networks consists of just product pooling and $1 \times 1$ *conv* layers. Having no point-wise activations between the layers, the non-linearity of the models stems from the product pooling operation itself. The pooling layers also control the depth of the network by the choice of the size and the shape of pooling windows. The *conv* operator is not unlike the standard convolutional layer of ConvNets, with the sole difference being that it may operate without *coefficient sharing*, i.e. the filters that generate feature maps by sliding across the

previous layer may have different coefficients at different spatial locations. This is often referred to in the deep learning community as a locally-connected operator (Taigman et al., 2014).

Arithmetic Circuits constructed from the above conv and product pooling layers are called *Convolutional Arithmetic Circuits*, or ConvACs for short, first suggested by Cohen et al. (2016a) as a theoretical framework for studying standard convolutional networks, sharing many of the defining traits of the latter, most noteworthy, the locality, sharing and pooling properties of ConvNets. Unlike general circuits, the structure of the network is determined solely by two parameters, the number of channels of each conv layer and the size of pooling windows, which indirectly controls the depth of the network. Any decomposition that corresponds to a ConvAC can represent any tensor, given sufficient number of channels, though deeper circuits result in more efficient representations (Cohen et al., 2016a).

Finally, since we are dealing with generative models, the tensors we study are non-negative and sum to one, i.e. the vectorization of $\mathcal{A}$ (rearranging its entries to the shape of a vector), denoted by $\text{vec}(\mathcal{A})$, is constrained to lie in the multi-dimensional simplex, denoted by:

$$\triangle^k := \left\{ \mathbf{x} \in \mathbb{R}^{k+1} | \sum_{i=1}^{k+1} x_i = 1, \forall i \in [k+1] : x_i \geq 0 \right\} \tag{1}$$

## 3 TENSORIAL MIXTURE MODELS

We represent the input signal $X$ by a sequence of low-dimensional *local structures*

$$X = (\mathbf{x}_1, \ldots, \mathbf{x}_N) \in (\mathbb{R}^s)^N$$

This representation is quite natural for many high-dimensional input domains such as images – where the local structures represent patches consisting of $s$ pixels – voice through spectrograms, and text through words.

A well-known observation, which has been verified in several empirical studies (e.g. by Zoran and Weiss (2011)), is that the distributions of local structures typically found in natural data could be sufficiently modeled by a mixture model consisting of only few components (on the order of 100) of simple distributions (e.g. Gaussian). Assuming the above holds for $X \in (\mathbb{R}^s)^N$ and let $\{P(\mathbf{x}|d; \theta_d)\}_{d=1}^M$ be the mixing components, parameterized by $\theta_1, \ldots, \theta_M$, from which local structures are generated, i.e. for all $i \in [N]$ there exist $d_i \in [M]$ such that $\mathbf{x}_i \sim P(\mathbf{x}|d_i; \theta_{d_i})$, where $d_i$ is a hidden variable specifying the matching component for the $i$-th local structure, then the probability density of sampling $X$ is fully described by:

$$P(X) = \sum_{d_1, \ldots, d_N = 1}^{M} P(d_1, \ldots, d_N) \prod_{i=1}^{N} P(\mathbf{x}_i | d_i; \theta_{d_i}) \tag{2}$$

where $P(d_1, \ldots, d_N)$ represents the prior probability of assigning components $d_1, \ldots, d_N$ to their respective local structures $\mathbf{x}_1, \ldots, \mathbf{x}_N$. Even though we had to make an assumption on $X$ to derive eq. 2, it is important to note that if we allow $M$ to become unbounded, then any distribution with support in $(\mathbb{R}^s)^N$ could be approximated by this equation. The argument follows from the universality property of the common parametric families of distributions (Gaussian, Laplacian, etc.), where any distribution can be approximated given sufficient number of components from these families, and thus the assumption always holds to some degree (see app. B for the complete proof).

The prior probabilities $P(d_1, \ldots, d_N)$ can also be represented by a tensor $\mathcal{A} \in (\mathbb{R}^M)^{\otimes N}$ of order $N$, given that the vectorization of $\mathcal{A}$ is constrained to the simplex, i.e. $\text{vec}(\mathcal{A}) \in \triangle^{(M^N - 1)}$ (see eq. 1). Thus, we refer to eq. 2 as a *Tensorial Mixture Model* (TMM) with *priors tensor* $\mathcal{A}$ and *mixing components* $P(\mathbf{x}|d_1; \theta_1), \ldots, P(\mathbf{x}|d_N; \theta_N)$. Notice that if $N = 1$ then we obtain the standard mixture model, whereas for a general $N$ it is equivalent to a mixture model with tensorised mixing weights and conditionally independent mixing components.

Unlike standard mixture models, we cannot perform inference directly from eq. 2, nor can we even store the priors tensor directly given its exponential size of $M^N$ entries. Therefore the TMM as presented by eq. 2 is *not tractable*. The way to make the TMM tractable is to replace the tensor $\mathcal{A}_{d_1, \ldots, d_N}$ by a tensor decomposition and, as described in the previous section, this gives rise to arithmetic circuits. But before we present our approach for tractable TMMs through tensor decompositions, it is worth examining some of the TMM special cases and how they relate to other known generative models.

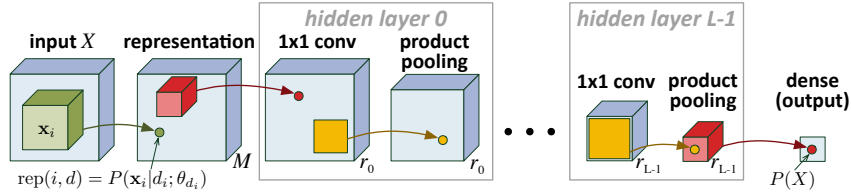

Figure 2: Inference of a TMM carried out by a ConvAC.

## 3.1 SPECIAL CASES

We have already shown that TMMs can be thought of as a special case of mixture models, but it is important to also note that diagonal Gaussian Mixture Models (GMMs), probably the most common type of mixture models, are a strict subset of TMMs. Assume $M = N \cdot K$, as well as:

$$P(d_1, \dots, d_N) = \begin{cases} w_k & \forall i \in [N], \ d_i = N \cdot (k-1) + i \\ 0 & \text{Otherwise} \end{cases}$$

$$P(\mathbf{x}|d; \theta_d) = \mathcal{N}(\mathbf{x}; \boldsymbol{\mu}_{ki}, \text{diag}(\boldsymbol{\sigma}_{ki}^2)), \ d = N \cdot (k-1) + i$$

then eq. 2 reduces to:

$$P(X) = \sum_{k=1}^{K} w_k \prod_{i=1}^{N} \mathcal{N}(\mathbf{x}; \boldsymbol{\mu}_{ki}, \text{diag}(\boldsymbol{\sigma}_{ki}^2)) = \sum_{k=1}^{K} w_k \mathcal{N}(\mathbf{x}; \tilde{\boldsymbol{\mu}}_k, \text{diag}(\tilde{\boldsymbol{\sigma}}_k^2))$$
$$\tilde{\boldsymbol{\mu}}_k = (\boldsymbol{\mu}_{k1}^T, \dots, \boldsymbol{\mu}_{kN}^T)^T \qquad \tilde{\boldsymbol{\sigma}}_k^2 = ((\boldsymbol{\sigma}_{k1}^2)^T, \dots, (\boldsymbol{\sigma}_{kN}^2)^T)^T$$

which is equivalent to a diagonal GMM with mixing weights $\mathbf{w} \in \triangle^{K-1}$ and Gaussian mixture components with means $\{\tilde{\boldsymbol{\mu}}_k\}_{k=1}^{K}$ and covariances $\{\text{diag}(\tilde{\boldsymbol{\sigma}}_k^2)\}_{k=1}^{K}$.

While the previous example highlights another connection between TMMs and mixture models, it does not take full advantage of the priors tensor, setting most of its entries to zero. Perhaps the simplest assumption we could make about the priors tensor, without it becoming degenerate, would be to assume that that the hidden variables $d_1, \dots, d_N$ are statistically independent, i.e. $P(d_1, \dots, d_N) = \prod_{i=1}^{N} P(d_i)$. Then rearranging eq. 2 will result in a product of mixture models:

$$P(X) = \prod_{i=1}^{N} \sum_{d=1}^{M} P(d_i = d) P(\mathbf{x}_i | d_i = d; \theta_d)$$

If we also assume that the priors are identical in addition to being independent, i.e. $P(d_1 = d) = \dots = P(d_N = d)$, then this model becomes a bag-of-words model, where the components $\{P(\mathbf{x}|d; \theta_d)\}_{d=1}^{M}$ define a soft dictionary for translating local-structures into "words", as is often done when applying bag-of-words models to images. Despite this familiar setting, had we subscribed to only using independent priors, we would lose the universality property of the general TMM model – it would not be capable of modeling dependencies between the local-structures.

## 3.2 DECOMPOSING THE PRIORS TENSOR

We have just seen that TMMs could be made tractable through constraints on the priors tensor, but it was at the expense of either not taking advantage of its tensor structure, or losing its universality property. Our approach for tractable TMMs is to apply tensor decompositions to the priors tensor, which is the conventional method for tackling the exponential size of high-order tensors.

We have already mentioned in sec. 2 that any decomposition representable by ConvACs, including the well-known CP and HT decompositions, can represent any tensor, and thus applying them would not limit the expressivity of our model. Fixing a ConvAC representing the priors tensor, i.e. $\Phi_\Theta(\delta_1, \dots, \delta_N) = \mathcal{A}_{d_1, \dots, d_N}$ where $\Theta$ are the parameters of the ConvAC and $\{\delta_i\}_{i=1}^{N}$ are the indicators representation of $\{d_i\}_{i=1}^{N}$, and simply rearranging the terms of eq. 2 after substituting the entries of the priors tensor with the sums and products expression of $\Phi_\Theta(\delta_1, \dots, \delta_N)$ results in:

$$P(X) = \Phi_\Theta(\mathbf{q}^1, \dots, \mathbf{q}^N) \qquad \forall i \in [N] \forall d \in [M], q_d^i = P(\mathbf{x}_i | d_i = d) \tag{3}$$

which is nearly equivalent to how the ConvAC is used for computing the entries of the priors tensor, differing only in the way the input vectors are defined. Namely, eq. 3 is a result of

replacing indicator vectors $\delta_i$ with probability vectors $\mathbf{q}^i$, which could be interpreted as a soft variant of indicator vectors. Viewed as a network, it begins with a *representation layer*, mapping the local structures to the likelihood probabilities of belonging to each mixing component, i.e. $\{\mathbf{x}_i\}_{i=1}^{N} \rightarrow \{P(\mathbf{x}_i|d_i{=}d;\theta_d)\}_{i=1,d=1}^{N,M}$. Following the representation layer is the same ConvAC described by $\Phi_\Theta(\cdot, \ldots, \cdot)$. The complete network is illustrated by fig. 2.

Unlike general tensors, for a TMM to represent a valid distribution, the priors tensor is constrained to the simplex and thus not every choice of parameters for the decomposition would result in a tensor holding this constraint. By restricting ourselves to non-negative decomposition parameters, i.e. use positive weights in the $1{\times}1$ *conv* layers, it guarantees the resulting tensors would be non-negative as well. Additionally, normalizing the non-negative tensor is equivalent to requiring the parameters to be restricted to the simplex, i.e. for every layer $l$ and spatial position $j$ the weight vector $\mathbf{w}^{l,j} \in \triangle^{r_{l-1}-1}$ of the respective $1{\times}1$ *conv* kernel is normalized to sum to one. Under these constraints we refer to it as a generative decomposition. Notice that restricting ourselves to generative decompositions does not limit the expressivity of our model, as we can still represent any non-negative tensor and thus any distribution that the original TMM could represent. In discussing the above, it helps to distinguish between the two extreme cases of generative decompositions representable by ConvACs, namely, the shallow Generative CP decomposition referred to as the *GCP-model*, and the deep Generative HT decomposition referred to as the *GHT-model*.

Non-negative matrix and tensor decompositions have a long history together with the development of corresponding generative models, e.g., pLSA (Hofmann, 1999) which uses non-negative matrix decompositions for text analysis, which was later extended for images with the help of "visual words" (Li and Perona, 2005). The non-negative variant of the CP decomposition presented above is related to the more general Latent Class Models (Zhang, 2004), which could be seen as a multi-dimensional pLSA. Likewise, the non-negative HT decomposition is related to the Latent Tree Model (Zhang, 2004; Mourad et al., 2013) with the structure of a complete binary tree. Thus both the GCP and GHT models can be represented as a two-level graphical model, where the top level is either an LCM or an LTM, and the bottom level represent the local structures which are conditionally sampled from the mixing components of the TMM.

To conclude, the application of ConvACs to decompose the priors tensor leads to tractable TMMs with inference implemented by convolutional networks, has deep roots to classical use of non-negative factorizations of generative models, and given sufficient resources does not limit expressivity. However, practical considerations raise the question on the extent of the expressive capacity of our models when the size of the ConvAC is polynomial with respect to the number of local structures and mixing components. This question was thoroughly studied in a series of works analyzing the importance of depth (Cohen et al., 2016a), compared them to the expressive capacity of ConvNets (Cohen and Shashua, 2016a), showing the latter is less capable than ConvACs, and the ability of ConvACs to model the dependency structure typically found in natural data (Cohen and Shashua, 2016b). We prove in app. D that their main results are not hindered by the introduction of simplex constraints to ConvACs as we did above. Together these results give us a detailed understanding of how the number of channels and size of pooling windows control the expressivity of the model. A more in depth overview of their results and its application to our models can be found in app. C.

### 3.3 COMPARISON TO SUM-PRODUCT NETWORKS

Sum-Product Networks (SPNs) are a related class of generative models which are also realized by Arithmetic Circuits, though not strictly convolutional circuits as defined above. While SPNs can realize any ConvAC and thus are universal and posses tractable inference, their lack of structure puts them at a disadvantage.

Picking the right SPN structure from the infinite possible combinations of sum and product nodes could be perplexing even for experts in the field. Indeed Poon and Domingos (2011); Gens and Domingos (2012) had to hand-engineer complex structures for each dataset guided by prior knowledge and heuristics, and while their results were impressive for their time, they are poor by current measures. This lead to many works studying the task of learning the structure directly from the data itself (Peharz et al., 2013; Gens and Domingos, 2013; Adel et al., 2015; Rooshenas and Lowd, 2014), which indeed improved upon manually designed SPNs on some tasks. Nevertheless, when

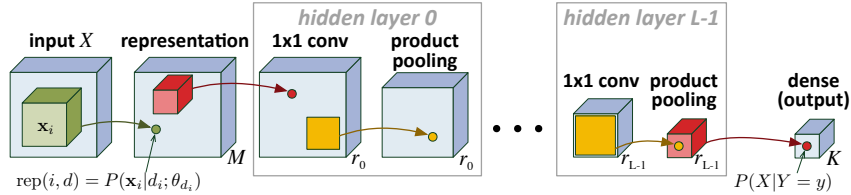

Figure 3: Classifier variant of TMM carried out by a ConvAC.

compared in absolute terms compared to other models, and not just average log-likelihood, they do not perform well even on simple handwritten digit classification datasets (Adel et al., 2015).

As opposed to SPNs, TMMs implemented with ConvACs have an easily designed architecture with only two set of parameters, size of pooling windows and number of channels, both of which can be directly related to the expressivity of the model as detailed in app. C. Additionally, while SPNs are typically trained using special EM-type algorithms, TMMs are trained using the stochastic gradient descent type algorithms as is common in training neural networks (see sec. 4 for details), thereby benefiting from the shared experience of a large and growing community.

## 4 CLASSIFICATION AND LEARNING WITH TMMS

Until this point we presented the TMM as a generative model for high-dimensional data, which is universal, and whose structure is tightly coupled to that of convolutional networks. We have yet to incorporate classification and learning into our framework. This is the purpose of the current section.

The common way to introduce object classes into a generative framework is to consider a class variable $Y$, and the distributions $P(X|Y)$ of the instance $X$ conditioned on $Y$. Under our model this is equivalent to having shared mixing components, but different priors tensors $P(d_1, \ldots, d_N | Y = y)$ for each class. Though it is possible to decompose each priors tensor separately, it is much more efficient to employ the concept of *joint tensor decomposition*, and use a shared ConvAC instead. This results in a single ConvAC computing inference, where instead of a single scalar output, multiple outputs are driven by the network – one for each class – as illustrated through the network in fig. 3.

Heading on to predicting the class of a given instance, we note that in practice, naïve implementation of ConvACs is not numerically stable, the reason being that high degree polynomials (as computed by such networks) are easily susceptible to numerical underflow or overflow. The conventional method for tackling this issue is to perform all computations in log-space. This transforms ConvACs into *SimNets*, a recently introduced deep learning architecture (Cohen and Shashua, 2014; Cohen et al., 2016b). Finally, prediction is carried by returning the most likely class, which in the common setting of uniform class priors ($P_\Theta(Y = y) \equiv 1/K$), translates to simply predicting the class for which the corresponding network output is maximal, in accordance with standard neural network practice:

$$\hat{Y}(X) = \operatorname{argmax}_y P(Y = y | X) = \operatorname{argmax}_y \log P(X | Y = y)$$

Suppose now that we are given a training set $S = \{(X^{(i)} \in (\mathbb{R}^s)^N, Y^{(i)} \in [K])\}_{i=1}^{|S|}$ of instances and labels, and would like to fit the parameters $\Theta$ of multi-class TMM according to the Maximum Likelihood method. Equivalently, we minimize the Negative Log-Likelihood (NLL) loss function: $\mathcal{L}(\Theta) = \mathbf{E}[-\log P_\Theta(X, Y)]$, which can be factorized into two separate loss functions:

$$\mathcal{L}(\Theta) = \mathbf{E}[-\log P_\Theta(Y | X)] + \mathbf{E}[-\log P_\Theta(X)]$$

where $\mathbf{E}[-\log P_\Theta(Y | X)]$ is commonly known as the cross-entropy loss, which we refer to as the *discriminative loss*, while $\mathbf{E}[-\log P_\Theta(X)]$ corresponds to maximizing the prior likelihood $P(X)$, and has no analogy in standard discriminative neural networks. It is this term that captures the generative nature of our model, and we accordingly refer to it as the *generative loss*. Now, let $N_\Theta(X^{(i)}; y) := \log P_\Theta(X^{(i)} | Y = y)$ stand for the $y$'th output of the SimNet (ConvAC in log-space) realizing the TMM with parameters $\Theta$, then in the case of uniform class priors, the empirical estimation of $\mathcal{L}(\Theta)$ may be written as:

$$\mathcal{L}(\Theta; S) = -\frac{1}{|S|} \sum_{i=1}^{|S|} \log \frac{e^{N_\Theta(X^{(i)}; Y^{(i)})}}{\sum_{y=1}^K e^{N_\Theta(X^{(i)}; y)}} - \frac{1}{|S|} \sum_{i=1}^{|S|} \log \sum_{y=1}^K e^{N_\Theta(X^{(i)}; y)} \qquad (4)$$

Maximum likelihood training of generative models is oftentimes based on dedicated algorithms such as Expectation-Maximization, which are typically difficult to apply at scale. We leverage the resemblance between our objective (eq. 4) and that of standard neural networks, and apply the same optimization procedures used for the latter, which have proven to be extremely effective for training classifiers at scale. Whereas other works have used tensor decompositions for the optimization of probabilistic models (Song et al., 2013; Anandkumar et al., 2014), we employ them strictly for modeling and instead make use of conventional methods. In particular, our implementation of TMMs is based on the SimNets extension of Caffe toolbox (Cohen et al., 2016b; Jia et al., 2014), and uses standard Stochastic Gradient Descent-type methods for optimization (see sec. 6 for more details).

## 5    CLASSIFICATION WITH MISSING DATA THROUGH MARGINALIZATION

A major advantage of generative models over discriminative ones lies in the ability to cope with missing data, specifically in the context of classification. By and large, discriminative methods either attempt to complete missing parts of the data before classification, known as *data imputation*, or learn directly to classify data with missing values (Little and Rubin, 2002). The first of these approaches relies on the quality of data completion, a much more difficult task than the original one of classification with missing data. Even if the completion was optimal, the resulting classifier is known to be sub-optimal (see app. E). The second approach does not make this assumption, but nonetheless assumes that the distribution of missing values at train and test times are similar, a condition which often does not hold in practice. Indeed, Globerson and Roweis (2006) coined the term "nightmare at test time" to refer to the common situation where a classifier must cope with missing data whose distribution is different from that encountered in training.

As opposed to discriminative methods, generative models are endowed with a natural mechanism for classification with missing data. Namely, a generative model can simply marginalize over missing values, effectively classifying under all possible completions, weighing each completion according to its probability. This, however, requires tractable inference and marginalization. We have already shown in sec. 3 that TMM support the former, and will show in sec. 5.1 bring forth marginalization which is just as efficient. Beforehand, we lay out the formulation of classification with missing data.

Let $\mathcal{X}$ be a random vector in $\mathbb{R}^s$ representing an object, and $\mathcal{Y}$ be a random variable in $[K]:=\{1,\ldots,K\}$ representing its label. Denote by $\mathcal{D}(\mathcal{X},\mathcal{Y})$ the joint distribution of $(\mathcal{X},\mathcal{Y})$, and by $(\mathbf{x}\in\mathbb{R}^s, y\in[K])$ specific realizations thereof. Assume that after sampling a specific instance $(\mathbf{x},y)$, a random binary vector $\mathcal{M}$ is drawn conditioned on $\mathcal{X}=\mathbf{x}$. More concretely, we sample a binary mask $\mathbf{m}\in\{0,1\}^s$ (realization of $\mathcal{M}$) according to a distribution $\mathcal{Q}(\cdot|\mathcal{X}=\mathbf{x})$. $x_i$ is considered missing if $m_i$ is equal to zero, and observed otherwise. Formally, we consider the vector $\mathbf{x}\odot\mathbf{m}$, whose $i$'th coordinate is defined to hold $x_i$ if $m_i=1$, and the wildcard $*$ if $m_i=0$. The classification task is then to predict $y$ given access solely to $\mathbf{x}\odot\mathbf{m}$.

Following the works of Rubin (1976); Little and Rubin (2002), we consider three cases for the missingness distribution $\mathcal{Q}(\mathcal{M}=\mathbf{m}|\mathcal{X}=\mathbf{x})$: missing completely at random (*MCAR*), where $\mathcal{M}$ is independent of $\mathcal{X}$, i.e. $\mathcal{Q}(\mathcal{M}=\mathbf{m}|\mathcal{X}=\mathbf{x})$ is a function of $\mathbf{m}$ but not of $\mathbf{x}$; missing at random (*MAR*), where $\mathcal{M}$ is independent of the missing values in $\mathcal{X}$, i.e. $\mathcal{Q}(\mathcal{M}=\mathbf{m}|\mathcal{X}=\mathbf{x})$ is a function of both $\mathbf{m}$ and $\mathbf{x}$, but is not affected by changes in $x_i$ if $m_i=0$; and missing not at random (*MNAR*), covering the rest of the distributions for which $\mathcal{M}$ depends on missing values in $\mathcal{X}$, i.e. $\mathcal{Q}(\mathcal{M}=\mathbf{m}|\mathcal{X}=\mathbf{x})$ is a function of both $\mathbf{m}$ and $\mathbf{x}$, which at least sometimes is sensitive to changes in $x_i$ when $m_i=0$.

Let $\mathcal{P}$ be the joint distribution of the object $\mathcal{X}$, label $\mathcal{Y}$, and missingness mask $\mathcal{M}$:

$$\mathcal{P}(\mathcal{X}=\mathbf{x}, \mathcal{Y}=y, \mathcal{M}=\mathbf{m}) = \mathcal{D}\left(\mathcal{X}=\mathbf{x}, \mathcal{Y}=y\right) \cdot \mathcal{Q}(\mathcal{M}=\mathbf{m}|\mathcal{X}=\mathbf{x})$$

For given $x \in \mathbb{R}^s$ and $\mathbf{m} \in \{0,1\}^s$, denote by $o(\mathbf{x}, \mathbf{m})$ the event where the random vector $\mathcal{X}$ coincides with $\mathbf{x}$ on the coordinates $i$ for which $m_i = 1$. For example, if $\mathbf{m}$ is an all-zero vector $o(\mathbf{x}, \mathbf{m})$ covers the entire probability space, and if $\mathbf{m}$ is an all-one vector $o(\mathbf{x}, \mathbf{m})$ corresponds to the event $\mathcal{X} = \mathbf{x}$. With these notations in hand, we are now in a position to characterize the optimal predictor in the presence of missing data:

**Claim 1.** *For any data distribution $\mathcal{D}$ and missingness distribution $\mathcal{Q}$, the optimal classification rule in terms of 0-1 loss is given by:*

$$h^*(\mathbf{x}\odot\mathbf{m}) = \mathrm{argmax}_y\, \mathcal{P}(\mathcal{Y}=y|o(\mathbf{x},\mathbf{m}))\mathcal{P}(\mathcal{M}=\mathbf{m}|o(\mathbf{x},\mathbf{m}),\mathcal{Y}=y)$$

*Proof.* See app. E. □

When the distribution $\mathcal{Q}$ is MAR (or MCAR), the classifier admits a simpler form, referred to as the *marginalized Bayes predictor*:

**Corollary 1.** *Under the conditions of claim 1, if the distribution $\mathcal{Q}$ is MAR (or MCAR), the optimal classification rule may be written as:*

$$h^*(\mathbf{x} \odot \mathbf{m}) = \operatorname{argmax}_y \ \mathcal{P}(\mathcal{Y}{=}y|o(\mathbf{x}, \mathbf{m})) \tag{5}$$

*Proof.* See app. E. □

Corollary 1 indicates that in the MAR setting, which is frequently encountered in practice, optimal classification does *not* require prior knowledge regarding the missingness distribution $\mathcal{Q}$. As long as one is able to realize the marginalized Bayes predictor (eq. 5), or equivalently, to compute the likelihoods of observed values conditioned on labels ($\mathcal{P}(o(\mathbf{x}, \mathbf{m})|Y{=}y)$), classification with missing data is guaranteed to be optimal, regardless of the corruption process taking place. This is in stark contrast to discriminative methods, which require access to the missingness distribution during training, and thus are not able to cope with unknown conditions at test time.

Most of this section dealt with the task of prediction given an input with missing data, where we assumed we had access to a complete and uncorrupted training set, and only faced missingness during prediction. However, many times we wish to tackle the reverse problem, where the training set itself is riddled with missing data. Generative methods can once again leverage their natural ability to handle missing data in the form of marginalization during the learning stage. Generative models are typically learned through the Maximum Likelihood principle. When it comes to learning from missing data, the marginalized likelihood objective is used instead. Under the MAR assumption, this method results in an unbiased classifier (Little and Rubin, 2002).

## 5.1 Efficient Marginalization with TMMs

As discussed above, with generative models optimal classification with missing data (in the MAR setting) is oblivious to the specific missingness distribution. However, it requires tractable computation of the likelihood of observed values conditioned on labels, i.e. tractable marginalization over missing values. The plurality of generative models that have recently gained attention in the deep learning community (Goodfellow et al., 2014; Kingma and Welling, 2014; Dinh et al., 2014; 2016) do not meet this requirement, and thus are not suitable for classification with missing data. TMMs on the other hand bring forth extremely efficient marginalization, requiring only a single forward pass through the corresponding network. Details follow.

Recall from sec. 3 and 4 that a multi-class TMM realizes the following form:

$$P(\mathbf{x}_1, \ldots, \mathbf{x}_N|Y{=}y) = \sum_{d_1, \ldots, d_N}^{M} P(d_1, \ldots, d_N|Y{=}y) \prod_{i=1}^{N} P(\mathbf{x}_i|d_i; \theta_{d_i}) \tag{6}$$

Suppose now that only the local structures $\mathbf{x}_{i_1} \ldots \mathbf{x}_{i_V}$ are observed, and we would like to marginalize over the rest. Integrating eq. 6 gives:

$$P(\mathbf{x}_{i_1}, \ldots, \mathbf{x}_{i_V}|Y{=}y) = \sum_{d_1, \ldots, d_N}^{M} P(d_1, \ldots, d_N|Y{=}y) \prod_{v=1}^{V} P(\mathbf{x}_{i_v}|d_{i_v}; \theta_{d_{i_v}})$$

from which it is evident that the same ConvAC used to compute $P(\mathbf{x}_1, \ldots, \mathbf{x}_N|Y{=}y)$, can be used to compute $P(\mathbf{x}_{i_1}, \ldots, \mathbf{x}_{i_V}|Y{=}y)$ – all it requires is a slight adaptation of the representation layer. Namely, the latter would represent observed values through the usual likelihoods, whereas missing (marginalized) values would now be represented via constant ones:

$$\operatorname{rep}(i, d) = \begin{cases} 1 & , \mathbf{x}_i \text{ is missing (marginalized)} \\ P(\mathbf{x}_i|d; \Theta) & , \mathbf{x}_i \text{ is visible (not marginalized)} \end{cases}$$

To conclude, with TMMs marginalizing over missing values is just as efficient as plain inference – requires only a single pass through the corresponding ConvAC. Accordingly, the marginalized Bayes predictor (eq. 5) is realized efficiently, and classification with missing data (in the MAR setting) is optimal, regardless of the missingness distribution. This capability is not provided by discriminative methods, which rely on the distribution of missing values being know at training, and by contemporary generative models, which do not bring forth tractable marginalization.

| | $N = 0$ | 25 | 50 | 75 | 100 | 125 | 150 |
|---|---|---|---|---|---|---|---|
| LP-Based | 97.9 | 97.5 | 96.4 | 94.1 | 89.2 | 80.9 | 70.2 |
| GHT-model | **98.5** | **98.2** | **97.8** | **96.5** | **93.9** | **87.1** | **76.3** |

Table 1: Blind classification with missing data on the binary MNIST dataset with feature deletion noise according to Globerson and Roweis (2006), averaged over all pairs of digits.

## 6 EXPERIMENTS

We demonstrate the properties of our models through both qualitative and quantitative experiments. In subsec. 6.1 we present our state-of-the-art results on image classification with missing data, with robustness to various missingness distributions. In app. G we show visualizations produced by our models, which gives us insight into its inner workings. Our experiments were conducted on the MNIST digit classification dataset, consisting of 60000 grayscale images of single digit numbers, as well as the small NORB 3D object recognition dataset, consisting of 48600 grayscale stereo images of toys belonging to 5 categories: four-legged animals, human figures, airplanes, trucks, and cars

In all our experiments we use either the GCP or GHT model with Gaussian mixing components. The weights of the conv layers are partially shared as described in sec 3.2, and are represented in log-space. For the case of the GHT model, we use $2 \times 2$ pooling windows for all pooling layers. We train our model according to the loss described in sec. 4, using the Adam (Kingma and Ba, 2015) variant of SGD and decaying learning rates. We apply $L^2$-regularization to the weights while taking into account they are stored in log-space. Additionally, we also adapt a probabilistic interpretation of dropout (**?**) by introducing random marginalization layers, that randomly select spatial locations in the input and marginalize over them. We provide a complete and detailed description of our experiments in app. F.

Our implementation, which is based on Caffe (Jia et al., 2014) and MAPS (Ben-Nun et al., 2015), as well as other code for reproducing our experiments, is available through our Github repository: `https://github.com/HUJI-Deep/TMM`.

### 6.1 IMAGE CLASSIFICATION WITH MISSING DATA

We demonstrate the effectiveness of our method for classification with missing data of unknown missingness distribution (see sec. 5), by conducting three kinds of experiments on the MNIST dataset, and an additional experiment on the NORB dataset. We begin by following the protocol of Globerson and Roweis (2006) – the binary classification problem of digit pairs with feature deletion noise – where we compare our method to the best known result on that benchmark (Dekel and Shamir, 2008). For our main experiment, we move to the harder multi-class digit classification under two different MAR missingness distributions, comparing against other methods which do not assume a specific missingness distribution. We repeat this experiment on the NORB dataset as well. Finally, our last experiment demonstrates the failure of purely discriminative methods to adapt to previously unseen missingness distributions, underlining the importance of the generative approach to missing data. We do wish to emphasize that missing data is not typically found in most image data, nevertheless, experiments on images with missing data are very common, for both classification and inpainting tasks. Additionally, there is nothing about our method, nor the methods we compare it against, that is very specific to the image domain, and thus any conclusion drawn should not be limited to the chosen datasets, but be taken in the broader context of the missing data problem.

The problem of learning classifiers which are robust to unforeseen missingness distributions at test time was first proposed by Globerson and Roweis (2006). They suggested missing values could be denoted by values which were *deleted*, i.e. their values were changed to zero, and a robust classifier would have to assume that any of its zero-value inputs could be the result of such a deletion process, and must be treated as missing. Their solution was to train a linear classifier and formulate the optimization as a quadric program under the constraint that $N$ of its features could be deleted. In Dekel and Shamir (2008), this solution was improved upon and generalized to other kinds of corruption beyond deletion as well as to an adversarial setting.

We follow the central experiment of these articles, conducted on binary classification of digits pairs from the MNIST dataset, where $N$ non-zero pixels are deleted with uniform probability over the set of $N$ non-zero pixel locations of the given image. We compare our method, using the deep GHT-

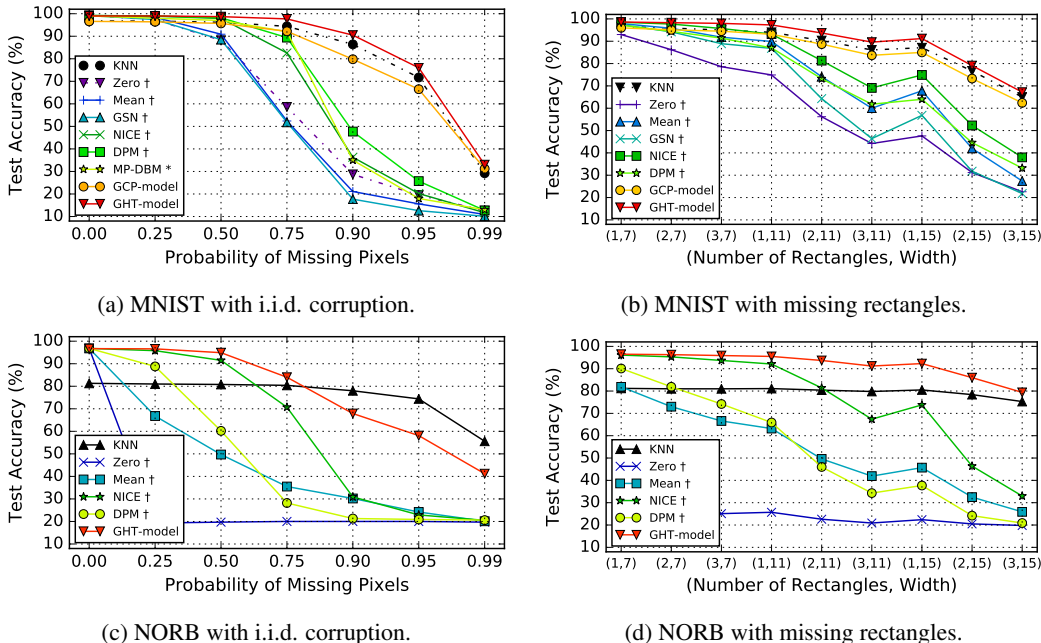

Figure 4: Blind classification with missing data. (a,c) Testing i.i.d. corruption with probability $p$ for each pixel. (b,d) Testing missing rectangles corruption with $N$ missing rectangles, each of width and hight equal to $W$. (*) Accuracies are estimated from the plot of Goodfellow et al. (2013). (†) Data imputation algorithms followed by a ConvNet. Raw results can be found in app. H.

Model, solely against the LP-based algorithm of Dekel and Shamir (2008), which is the previous state-of-the-art on this task. Due to the limited computational resources at the time, the original experiments were limited to training sets of just 50 images per digit. We have repeated their experiment, using the implementation kindly supplied to us by the authors, and increased the limit to 300 images per digit, which is the maximal amount possible with our current computational resources. Though it is possible to train our own models using much larger training sets, we have trained them under the same limitations. Despite the fact that missingness distribution of this experiment is of the MNAR type, which our method was not guarantied to be optimal under, the test results (see table 1) clearly show the large gap between our method and theirs. Additionally, whereas our method uses a single model trained once and with no prior knowledge on the missingness distribution, their method requires training special classifiers for each value of $N$, chosen through a cross-validation process, disqualifying it from being truly blind to the missingness distribution.

We continue to our main experiments on multi-class blind classification with missing data, where the missingness distribution is completely unknown during test time, and a single classifier must handle all possible distributions. We simulate two kinds of MAR missingness distributions: (i) an i.i.d. mask with a fixed probability $p \in [0, 1]$ of missing each pixel, and (ii) a mask composed of the union of $N$ possibly overlapping rectangles of width and height equal to $W$, each with a randomly assigned position in the image, distributed uniformly. We evaluate both our shallow GCP-Model as well as the deep GHT-Model against the most widely used methods for blind classification with missing data. We repeat these experiments on the MNIST and NORB datasets, the results of which are presented in fig. 4.

As a baseline for our results, we use K-Nearest Neighbors (KNN) to vote on the most likely class of a given example. We extend KNN to missing data by comparing distances using only the observed entries, i.e. for a corrupted instance $\mathbf{x} \odot \mathbf{m}$, and a clean image from the training set $\tilde{\mathbf{x}}$, we compute: $d(\tilde{\mathbf{x}}, \mathbf{x} \odot \mathbf{m}) = \sum_{m_{ij}=1} (\tilde{x}_{ij} - x_{ij})^2$. Though it scores better than the majority of modern methods we have compared, in practice KNN is very inefficient, even more so for missing data, which prevents most common memory and runtime optimizations typically employed to reduce its inefficiency. Additionally, KNN does not generalize well for more complex datasets, as is evident by its poor performance on the clean test set of the NORB dataset.

| $p_{\text{train}}$ \ $p_{\text{test}}$ | 0.25 | 0.50 | 0.75 | 0.90 | 0.95 | 0.99 |
|---|---|---|---|---|---|---|
| 0.25 | 98.9 | 97.8 | 78.9 | 32.4 | 17.6 | 11.0 |
| 0.50 | **99.1** | 98.6 | 94.6 | 68.1 | 37.9 | 12.9 |
| 0.75 | 98.9 | **98.7** | **97.2** | 83.9 | 56.4 | 16.7 |
| 0.90 | 97.6 | 97.5 | 96.7 | **89.0** | 71.0 | 21.3 |
| 0.95 | 95.7 | 95.6 | 94.8 | 88.3 | **74.0** | 30.5 |
| 0.99 | 87.3 | 86.7 | 85.0 | 78.2 | 66.2 | **31.3** |
| i.i.d. (rand) | 98.7 | 98.4 | 97.0 | 87.6 | 70.6 | 29.6 |
| rects (rand) | 98.2 | 95.7 | 83.2 | 54.7 | 35.8 | 17.5 |

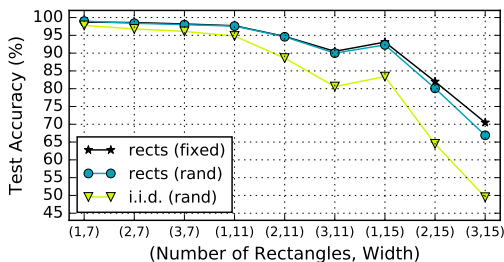

(a) MNIST with i.i.d. corruption (b) MNIST with missing rectangles.

Figure 5: We compare ConvNets trained on one distribution while tested on others. Training on randomly (rand) chosen distributions were also examined. (a) Trained on i.i.d. corruption with probability $p_{\text{train}}$, while tested on i.i.d. corruption with probability $p_{\text{test}}$. (b) Train and tested on the same (fixed) missing rectangles distribution, against ones trained on randomly chosen distributions.

As discusses in sec. 5, data-imputation is the most common method to handle missing data of unknown missingness distributions. Despite the popularity of this method, high quality data imputations are very hard to produce, amplified by the fact that classification algorithms are known to be highly sensitive to even a small noise applied to their inputs (**?**). Even if we assume the data-imputation step was done optimally, it would still not give optimal performance under all MAR missingness distributions, and under some settings could produce results which are only half as good as our method (see app. E for such a case). In our experiments, we have applied several data-imputations methods to complete the missing data, followed by classifying its outputs using a standard ConvNet fitted to the fully-observed training set. We first tested naive heuristics, filling missing values with zeros or the mean pixel value computed over all the images in the dataset. We then tested three generative models: GSN (Bengio et al., 2014), NICE (Dinh et al., 2014) and DPM (Sohl-Dickstein et al., 2015), which are known to work well for inpainting. GSN was omitted from the NORB experiments as we have not manage to properly train it on that dataset. Though the data-imputation methods are competitive when only few of the pixels are missing, they all fall far behind our models above a certain threshold, with more than 50 percentage points separating our GHT-model from the best data-imputation method under some of the cases. Additionally, all the generative models require very long runtimes, which prevents from using them in most real-world applications. While we tried to be as comprehensive as possible when choosing which inpainting methods to use, some of the most recent studies on the subject, e.g. the works of van den Oord et al. (2016) and Pathak et al. (2016), have either not yet published their code or only partially published it. We have also ruled out inpainting algorithms which are made specifically for images, as we did not want to limit the implications of these experiments solely to images.

We have also compared ourselves to the published results of the MPDBM model (Goodfellow et al., 2013). Unlike the previous generative models we tested, MPDBM is a generative classifier similar to our method. However, unlike our model, MPDBM does not posses the tractable marginalization nor the tractable inference properties, and uses approximations instead. Its lesser performance underlines the importance of these properties for achieving optimality under missing data. An additional factor might also be their training method, which includes randomly picking a subset of variables to act as missing, which might have introduced a bias to the specific missingness distribution used during their training.

In order to demonstrate the ineffectiveness of purely discriminative models, we trained ConvNets directly on randomly corrupted instances according to *pre-selected* missingness distributions on the MNIST dataset. Unlike the previous experiments, we do allow prior knowledge about the missingness distribution during training time. We found that the best results are achieved when replacing missing values with zeros, and adding as an extra input channel the mask of missing values (known as flag data-imputation). The results (see fig. 5) unequivocally show the effectiveness of this method when tested on the same distribution it was trained on, achieving a high accuracy even when only $10\%$ of the pixels are visible. However, when tested on different distributions, whether on a completely different kind or even on the same kind but with different parameters, the accuracy drops by a large factor, at times by more than 35 percentage points. This illustrate the disadvantage of the discriminative method, as it necessarily incorporates bias towards the corruption process it had seen during training, which makes it fail on other distributions. One might wonder whether it is

possible for a single network to be robust on more than a single distribution. We found out that the latter is true, and if we train a network on multiple different missingness distributions[1], then the network will achieve good performance on all such distributions, though at some cases not reaching the optimal performance. However, though it is possible to train a network to be robust on more than one distribution, the type of missingness distributions are rarely known in advance, and there is no known method to train a neural network against all possible distributions, limiting the effectivity of this method in practice.

Unlike all the above methods, our GHT-model, which is trained only once on the clean dataset, match or sometimes even surpass the performance of ConvNets that are trained and tested on the same distribution, showing it is achieving near optimal performance – as much as possible on any given distribution. Additionally, note that similar to ConvNets and according to the theory in app. C, the deep GHT-model is decidedly superior to the shallow GCP-model. Experimenting on more complex datasets is left for further research. Progress on optimization and regularization of networks based on product pooling (even in log-space) is required, and ways to incorporate larger $b \times b$ convolutional operations with overlaps would be useful before we venture into larger and complex datasets. Nevertheless, our preliminary results demonstrate an overwhelming advantage of our TMM models compared to competing methods, both in terms of robustness to different types of missing data, as well as in terms of raw performance, with very wide gaps in absolute accuracy than the next best method, at times as large as 50 percentage points more than the next best method.

## 7 SUMMARY

We have introduced a new family of probabilistic models, which we call Tensorial Mixture Models. TMMs are based on a simple assumption on the data, which stems from known empirical results on natural images, that gives rise to mixture models with tensorial structure represented by the *priors tensor*. When the priors tensor is decomposed it gives rise to an arithmetic circuit which in turn transforms the TMM into a Convolutional Arithmetic Circuit (ConvAC). A ConvAC corresponds to a shallow (single hidden layer) network when the priors tensor is decomposed by a CP (sum of rank-1) approach and corresponds to a deep network when the decomposition follows the Hierarchical Tucker (HT) model.

The ConvAC representation of a TMM possesses several attractive properties. First, the inference is tractable and is implemented by a forward pass through a deep network. Second, the architectural design of the model follows the deep networks community design, i.e., the structure of TMMs is determined by just two easily understood factors: size of pooling windows and number of channels. Finally, we have demonstrated the effectiveness of our model when tackling the problem of classification with missing data, leveraging TMMs unique ability of tractable marginalization which leads to optimal classifiers regardless of the missingness distribution.

There are several avenues for future research on TMMs which we are currently looking at, including other problems which TMMs could solve (e.g. semi-supervised learning), experimenting with other ConvACs architectures (e.g. through different decompositions), and further progress on optimization and regularization of networks with product pooling.

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

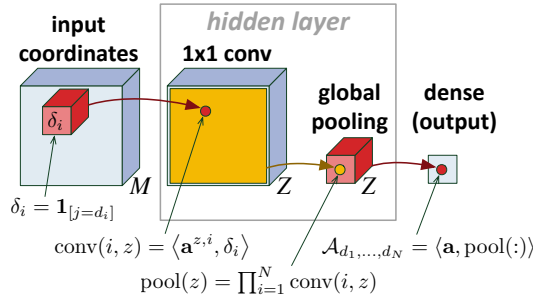

$$\delta_i = \mathbf{1}_{[j=d_i]}$$

$$\text{conv}(i,z) = \langle \mathbf{a}^{z,i}, \delta_i \rangle \qquad \mathcal{A}_{d_1,...,d_N} = \langle \mathbf{a}, \text{pool}(:) \rangle$$

$$\text{pool}(z) = \prod_{i=1}^{N} \text{conv}(i,z)$$

Figure 6: The decoding algorithm of the CP decomposition represented by an Arithmetic Circuit.

# A   BACKGROUND ON TENSOR DECOMPOSITIONS AND CONVOLUTIONAL ARITHMETIC CIRCUITS

We begin by establishing the minimal background in the field of tensor analysis required for following our work. A tensor is best thought of as a multi-dimensional array $\mathcal{A}_{d_1,...,d_N} \in \mathbb{R}$, where $\forall i \in [N], d_i \in [M_i]$. The number of indexing entries in the array, which are also called *modes*, is referred to as the *order* of the tensor. The number of values an index of a particular mode can take is referred to as the *dimension* of the mode. The tensor $\mathcal{A} \in \mathbb{R}^{M_1 \otimes \cdots \otimes M_N}$ mentioned above is thus of order $N$ with dimension $M_i$ in its $i$-th mode. For our purposes we typically assume that $M_1 = \ldots = M_N = M$, and simply denote it as $\mathcal{A} \in (\mathbb{R}^M)^{\otimes N}$.

The fundamental operator in tensor analysis is the *tensor product*. The tensor product operator, denoted by $\otimes$, is a generalization of outer product of vectors (1-ordered vectors) to any pair of tensors. Specifically, let $\mathcal{A}$ and $\mathcal{B}$ be tensors of order $P$ and $Q$ respectively, then the tensor product $\mathcal{A} \otimes \mathcal{B}$ results in a tensor of order $P + Q$, defined by: $(\mathcal{A} \otimes \mathcal{B})_{d_1,...,d_{P+Q}} = \mathcal{A}_{d_1,...,d_P} \cdot \mathcal{B}_{d_{P+1},...,d_{P+Q}}$.

The main concept from tensor analysis we use in our work is that of tensor decompositions. The most straightforward and common tensor decomposition format is the rank-1 decomposition, also known as a CANDE-COMP/PARAFAC decomposition, or in short, a *CP decomposition*. The CP decomposition is a natural extension of low-rank matrix decomposition to general tensors, both built upon the concept of a linear combination of rank-1 elements. Similarly to matrices, tensors of the form $\mathbf{v}^{(1)} \otimes \cdots \otimes \mathbf{v}^{(N)}$, where $\mathbf{v}^{(i)} \in \mathbb{R}^{M_i}$ are non-zero vectors, are regarded as $N$-ordered rank-1 tensors, thus the rank-$Z$ CP decomposition of a tensor $\mathcal{A}$ is naturally defined by:

$$\mathcal{A} = \sum_{z=1}^{Z} a_z \mathbf{a}^{z,1} \otimes \cdots \otimes \mathbf{a}^{z,N}$$

$$\Rightarrow \mathcal{A}_{d_1,...,d_N} = \sum_{z=1}^{Z} a_z \prod_{i=1}^{N} a_{d_i}^{z,i} \tag{7}$$

where $\{\mathbf{a}^{z,i} \in \mathbb{R}^{M_i}\}_{i=1,z=1}^{N,Z}$ and $\mathbf{a} \in \mathbb{R}^Z$ are the parameters of the decomposition. As mentioned above, for $N = 2$ it is equivalent to low-order matrix factorization. It is simple to show that any tensor $\mathcal{A}$ can be represented by the CP decomposition for some $Z$, where the minimal such $Z$ is known as its *tensor rank*.

Another decomposition we will use in this paper is of a hierarchical nature and known as the Hierarchical Tucker decomposition (Hackbusch and Kühn, 2009), which we will refer to as *HT decomposition*. While the CP decomposition combines vectors into higher order tensors in a single step, the HT decomposition does that more gradually, combining vectors into matrices, these matrices into 4th ordered tensors and so on recursively in a hierarchically fashion. Specifically, the following describes the recursive formula of the HT decomposition[2]

---

[2] More precisely, we use a special case of the canonical HT decomposition as presented in Hackbusch and Kühn (2009). In the terminology of the latter, the matrices $A^{l,j,\gamma}$ are diagonal and equal to $diag(\mathbf{a}^{l,j,\gamma})$ (using the notations from eq. 8).

for a tensor $\mathcal{A} \in (\mathbb{R}^M)^{\otimes N}$ where $N = 2^L$, i.e. $N$ is a power of two[3]:

$$\phi^{1,j,\gamma} = \sum_{\alpha=1}^{r_0} a_\alpha^{1,j,\gamma} \mathbf{a}^{0,2j-1,\alpha} \otimes \mathbf{a}^{0,2j,\alpha}$$

$$\cdots$$

$$\phi^{l,j,\gamma} = \sum_{\alpha=1}^{r_{l-1}} a_\alpha^{l,j,\gamma} \underbrace{\phi^{l-1,2j-1,\alpha}}_{\text{order } 2^{l-1}} \otimes \underbrace{\phi^{l-1,2j,\alpha}}_{\text{order } 2^{l-1}}$$

$$\cdots$$

$$\phi^{L-1,j,\gamma} = \sum_{\alpha=1}^{r_{L-2}} a_\alpha^{L-1,j,\gamma} \underbrace{\phi^{L-2,2j-1,\alpha}}_{\text{order } \frac{N}{4}} \otimes \underbrace{\phi^{L-2,2j,\alpha}}_{\text{order } \frac{N}{4}}$$

$$\mathcal{A} = \sum_{\alpha=1}^{r_{L-1}} a_\alpha^L \underbrace{\phi^{L-1,1,\alpha}}_{\text{order } \frac{N}{2}} \otimes \underbrace{\phi^{L-1,2,\alpha}}_{\text{order } \frac{N}{2}} \tag{8}$$

where the parameters of the decomposition are the vectors $\{\mathbf{a}^{l,j,\gamma} \in \mathbb{R}^{r_{l-1}}\}_{l \in \{0,\ldots,L-1\}, j \in [N/2^l], \gamma \in [r_l]}$ and the top level vector $\mathbf{a}^L \in \mathbb{R}^{r_{L-1}}$, and the scalars $r_0, \ldots, r_{L-1} \in \mathbb{N}$ are referred to as the *ranks of the decomposition*. Similar to the CP decomposition, any tensor can be represented by an HT decomposition. Moreover, any given CP decomposition can be converted to an HT decomposition by only a polynomial increase in the number of parameters.

The relationship between tensor decomposition and networks arises from the simple observation that through decomposition one can tradeoff storage complexity with computation where the type of computation consists of sums and products. Specifically, tensor decompositions could be seen as a mapping, that takes a tensor of exponential size and converts it into a polynomially sized representation, coupled with a decoding algorithm of polynomial runtime complexity to retrieve the original entries of tensor – essentially trading off space complexity for computational complexity. Examining the decoding algorithms for the CP and HT decompositions, i.e. eq. 7 and eq. 8, respectively, reveal a shared framework for representing these algorithms via computation graphs of products and weighted sums, also known as *Arithmetic Circuits* (Shpilka and Yehudayoff, 2010) or Sum-Product Networks (Poon and Domingos, 2011). More specifically, these circuits take as input $N$ indicator vectors $\delta_1, \ldots, \delta_N$, representing the coordinates $(d_1, \ldots, d_N)$, where $\delta_i = \mathbf{1}_{[j=d_i]}$, and output the value of $\mathcal{A}_{d_1,\ldots,d_N}$. In the case of the CP decomposition, the matching decoding circuit is defined by eq. 9 below:

$$a_{d_i}^{z,i} = \sum_{d=1}^M a_d^{z,i} \delta_{id} \quad \Rightarrow \quad \mathcal{A}_{d_1,\ldots,d_N} = \sum_{z=1}^Z a_z \prod_{i=1}^N \sum_{d=1}^M a_d^{z,i} \delta_{id} \tag{9}$$

The above formula is better represented by the network illustrated in fig. 6, beginning with an input layer of $\sqrt{N} \times \sqrt{N}$ $M$-dimensional indicator vectors arranged in a 3D array, followed by a $1 \times 1$ *conv* operator, a global product pooling layer, and ends with a dense linear layer outputting $\mathcal{A}_{d_1,\ldots,d_N}$. The *conv* operator is not unlike the standard convolutional layer of ConvNets, with the sole difference being that it may operate without *coefficient sharing*, i.e. the filters that generate feature maps by sliding across the previous layer may have different coefficients at different spatial locations. This is often referred to in the deep learning community as a locally-connected operator (Taigman et al., 2014). Similarly to the CP decomposition, retrieving the entries of a tensor from its HT decomposition can be computed by the circuit represented in fig. 7, where instead of a single pair of conv and pooling layers there are $\log_2 N$ such pairs, with pooling windows of size 2. Though the canonical HT decomposition dictates size 2 pooling windows, any pooling structure used in practice still results in a valid HT decomposition.

Arithmetic Circuits constructed from the above conv and product pooling layers are called *Convolutional Arithmetic Circuits*, or ConvACs for short, first suggested by Cohen et al. (2016a) as a theoretical framework for studying standard convolutional networks, sharing many of the defining traits of the latter, most noteworthy, the locality, sharing and pooling properties of ConvNets. Unlike general circuits, the structure of the network is determined solely by two parameters, the number of channels of each conv layer and the size of pooling windows, which indirectly controls the depth of the network.

---

[3]The requirement for $N$ to be a power of two is solely for simplifying the definition of the HT decomposition. More generally, instead of defining it through a complete binary tree describing the order of operations, the canonical decomposition can use any balanced binary tree.

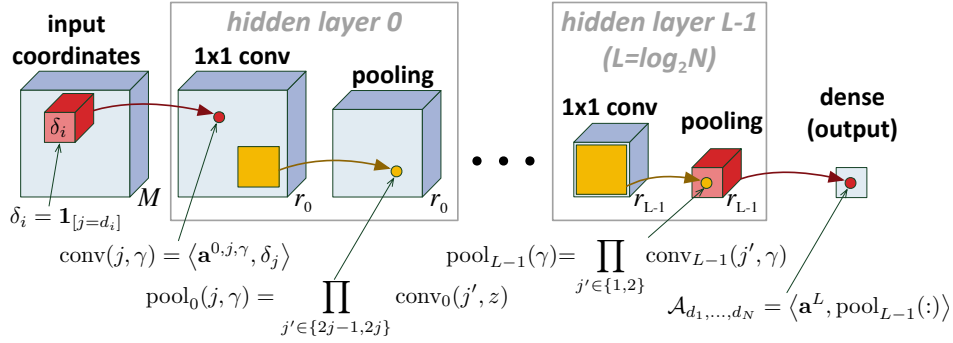

Figure 7: The decoding algorithm of the HT decomposition represented by an Arithmetic Circuit.

## B    THE UNIVERSALITY OF TENSORIAL MIXTURE MODELS

In this section we prove the universality property of TMMs, as discussed in sec. 3. We begin by taking note from functional analysis and define a new property called *PDF total set*, which is similar in concept to a *total set*, followed by proving that this property is invariant under the cartesian product of functions, which entails the universality of TMMs as a corollary.

**Definition 1.** *Let $\mathcal{F}$ be a set of PDFs over $\mathbb{R}^s$. $\mathcal{F}$ is PDF total iff for any PDF $h(\mathbf{x})$ over $\mathbb{R}^s$ and for all $\epsilon > 0$ there exists $M \in \mathbb{N}$, $\{f_1(\mathbf{x}), \ldots, f_M(\mathbf{x})\} \subset \mathcal{F}$ and $\mathbf{w} \in \triangle^{M-1}$ s.t. $\left\| h(\mathbf{x}) - \sum_{i=1}^{M} w_i f_i(\mathbf{x}) \right\|_1 < \epsilon$. In other words, a set is a PDF total set if its convex span is a dense set under $L^1$ norm.*

**Claim 2.** *Let $\mathcal{F}$ be a set of PDFs over $\mathbb{R}^s$ and let $\mathcal{F}^{\otimes N} = \{\prod_{i=1}^{N} f_i(\mathbf{x}) | \forall i, f_i(\mathbf{x}) \in \mathcal{F}\}$ be a set of PDFs over the product space $(\mathbb{R}^s)^N$. If $\mathcal{F}$ is a PDF total set then $\mathcal{F}^{\otimes N}$ is PDF total set.*

*Proof.* If $\mathcal{F}$ is the set of Gaussian PDFs over $\mathbb{R}^s$ with diagonal covariance matrices, which is known to be a PDF total set, then $\mathcal{F}^{\otimes N}$ is the set of Gaussian PDFs over $(\mathbb{R}^s)^N$ with diagonal covariance matrices and the claim is trivially true.

Otherwise, let $h(\mathbf{x}_1, \ldots, \mathbf{x}_N)$ be a PDF over $(\mathbb{R}^s)^N$ and let $\epsilon > 0$. From the above, there exists $K \in \mathbb{N}$, $\mathbf{w} \in \triangle^{M_1 - 1}$ and a set of diagonal Gaussians $\{g_{ij}(\mathbf{x})\}_{i \in [M_1], j \in [N]}$ s.t.

$$\left\| g(\mathbf{x}) - \sum_{i=1}^{M_1} w_i \prod_{j=1}^{N} g_{ij}(\mathbf{x}_j) \right\|_1 < \frac{\epsilon}{2} \tag{10}$$

Additionally, since $\mathcal{F}$ is a PDF total set then there exists $M_2 \in \mathbb{N}$, $\{f_k(\mathbf{x})\}_{k \in [M_2]} \subset \mathcal{F}$ and $\{\mathbf{w}_{ij} \in \triangle^{M_2 - 1}\}_{i \in [M_1], j \in [N]}$ s.t. for all $i \in [M_1], j \in [N]$ it holds that $\left\| g_{ij}(\mathbf{x}) - \sum_{k=1}^{M_2} w_{ijk} f_k(\mathbf{x}) \right\|_1 < \frac{\epsilon}{2N}$, from which it is trivially proven using a telescopic sum and the triangle inequality that:

$$\left\| \sum_{i=1}^{M_1} w_i \prod_{j=1}^{N} g_{ij}(\mathbf{x}) - \sum_{i=1}^{M_1} w_i \prod_{j=1}^{N} \sum_{k=1}^{M_2} w_{ijk} f_k(\mathbf{x}_j) \right\|_1 < \frac{\epsilon}{2} \tag{11}$$

From eq. 10, eq. 11 the triangle inequality it holds that:

$$\left\| g(\mathbf{x}) - \sum_{k_1, \ldots, k_N = 1}^{M_2} \mathcal{A}_{k_1, \ldots, k_N} \prod_{j=1}^{N} f_{k_j}(\mathbf{x}_j) \right\|_1 < \epsilon$$

where $\mathcal{A}_{k_1, \ldots, k_N} = \sum_{i=1}^{M_1} w_i \prod_{j=1}^{N} w_{ijk_j}$ which holds $\sum_{k_1, \ldots, k_N = 1}^{M_2} \mathcal{A}_{k_1, \ldots, k_N} = 1$. Taking $M = M_2^N$, $\{\prod_{j=1}^{N} f_{k_j}(\mathbf{x}_j)\}_{k_1 \in [M_2], \ldots, k_N \in [M_2]} \subset \mathcal{F}^{\otimes N}$ and $\mathbf{w} = \text{vec}(\mathcal{A})$ completes the proof. $\square$

**Corollary 2.** *Let $\mathcal{F}$ be a PDF total set of PDFs over $\mathbb{R}^s$, then the family of TMMs with mixture components from $\mathcal{F}$ can approximate any $PDF$ over $(\mathbb{R}^s)^N$ arbitrarily well, given arbitrarily many components.*

## C    OVERVIEW ON THE EXPRESSIVE CAPACITY OF CONVOLUTIONAL ARITHMETIC CIRCUITS AND ITS AFFECT ON TENSORIAL MIXTURE MODELS

The expressiveness of ConvACs has been extensively studied, and specifically the non-generative variants of our models, named CP-model and HT-model respectively. In Cohen et al. (2016a) it was shown that ConvACs

posses the property known as *complete depth efficiency*. Namely, almost all functions[4] realized by an HT-model of polynomial size, for them to be realized (or approximated) by a CP-model, require it to be of exponential size. In other words, the expressiveness borne out of depth is exponentially stronger than a shallow network, almost always. It is worth noting that in the followup paper (Cohen and Shashua, 2016a), the authors have shown that the same result does not hold for standard ConvNets – while there are specific instances where depth efficiency holds, it is not complete, i.e. there is a non-zero probability that a function realized by a polynomially sized deep ConvNet can also be realized by a polynomially sized shallow ConvNet. Despite the additional simplex constraints put on the parameters, complete depth efficiency does hold for the generative ConvACs of our work, proof of which can be found in app. D, which shows the advantage of the deeper GHT-model over the shallow GCP-model. Additionally, this illustrates how the two factors controlling the architecture – number of channels and size of pooling windows – control the expressive capacity of the GHT-model. While the above shows why the deeper GHT-model is preferred over the shallow GCP-model, there is still the question of whether a polynomially sized GHT-model is sufficient for describing the complexities of natural data. Though a complete and definite answer is unknown as of yet, there are some strong theoretical evidence that it might. One aspect of being sufficient for modeling natural data is the ability of the model to describe the dependency structures typically found in the data. In Cohen and Shashua (2016b), the authors studied the *separation rank* – a measure of correlation, which for a given input partition, measures how far a function is from being separable – and found that a polynomially sized HT-model is capable of exponential separation rank for interleaved partitions, i.e. that it can model high correlations in local areas in the input. Additionally, for non-contiguous partitions, the separation rank can be at most polynomial, i.e. it can only model a limited correlation between far away areas in the input. These two results combined suggest that the HT-model, and thus also our GHT-model, is especially fit for modeling the type of correlations typically found in natural images and audio, even if it is only of polynomial size. Finally, from an empirical perspective, convolutional hierarchical structures have shown great success on multitude of different domains and tasks. Our models leverage these structures, taking them to a probabilistic setting, which leads us to believe that they will be able to effectively model distributions in practice – a belief we verify by experiments.

## D  PROOF FOR THE DEPTH EFFICIENCY OF GENERATIVE CONVOLUTIONAL ARITHMETIC CIRCUITS

In this section we prove that the depth efficiency property of ConvACs proved in Cohen et al. (2016a) applies also to the Generative ConvACs we have introduced in sec. 3.2. More specifically, we prove the following theorem, which is the generative analog of theorem 1 from (Cohen et al., 2016a):

**Theorem 1.** *Let $\mathcal{A}^y$ be a tensor of order $N$ and dimension $M$ in each mode, generated by the recursive formulas in eq. 8, under the simplex constraints introduced in sec. 3.2. Define $r := \min\{r_0, M\}$, and consider the space of all possible configurations for the parameters of the decomposition – $\{\mathbf{a}^{l,j,\gamma} \in \triangle^{r_{l-1}-1}\}_{l,j,\gamma}$. In this space, the generated tensor $\mathcal{A}^y$ will have CP-rank of at least $r^{N/2}$ almost everywhere (w.r.t. the product measure of simplex spaces). Put differently, the configurations for which the CP-rank of $\mathcal{A}^y$ is less than $r^{N/2}$ form a set of measure zero. The exact same result holds if we constrain the composition to be "shared", i.e. set $\mathbf{a}^{l,j,\gamma} \equiv \mathbf{a}^{l,\gamma}$ and consider the space of $\{\mathbf{a}^{l,\gamma} \in \triangle^{r_{l-1}-1}\}_{l,\gamma}$ configurations.*

The only differences between ConvACs and their generative counter-parts are the simplex constraints applied to the parameters of the models, which necessitate a careful treatment to the measure theoretical arguments of the original proof. More specifically, while the $k$-dimensional simplex $\triangle^k$ is a subset of the $k+1$-dimensional space $\mathbb{R}^{k+1}$, it has a zero measure with respect to the Lebesgue measure over $\mathbb{R}^{k+1}$. The standard method to define a measure over $\triangle^k$ is by the Lebesgue measure over $\mathbb{R}^k$ of its projection to that space, i.e. let $\lambda : \mathbb{R}^k \to \mathbb{R}$ be the Lebesgue measure over $\mathbb{R}^k$, $p : \mathbb{R}^{k+1} \to \mathbb{R}^k, p(\mathbf{x}) = (x_1, \ldots, x_k)^T$ be a projection, and $A \subset \triangle^k$ be a subset of the simplex, then the latter's measure is defined as $\lambda(p(A))$. Notice that $p(\triangle^k)$ has a positive measure, and moreover that $p$ is invertible over the set $p(\triangle^k)$, and that its inverse is given by $p^{-1}(x_1, \ldots, x_k) = (x_1, \ldots, x_k, 1 - \sum_{i=1}^{k} x_i)$. In our case, the parameter space is the cartesian product of several simplex spaces of different dimensions, for each of them the measure is defined as above, and the measure over their cartesian product is uniquely defined by the product measure. Though standard, the choice of the projection function $p$ above could be seen as a limitation, however, the set of zero measure sets in $\triangle^k$ is identical for any reasonable choice of a projection $\pi$ (e.g. all polynomial mappings). More specifically, for any projection $\pi : \mathbb{R}^{k+1} \to \mathbb{R}^k$ that is invertible over $\pi(\triangle^k)$, $\pi^{-1}$ is differentiable, and the Jacobian of $\pi^{-1}$ is bounded over $\pi(\triangle^k)$, then a subset $A \subset \triangle^k$ is of measure zero w.r.t. the projection $\pi$ iff it is of measure zero w.r.t. $p$ (as defined above). This implies that if we sample the weights of the generative decomposition (eq. 8 with simplex constraints) by a continuous distribution, a property that holds with probability 1 under the standard parameterization (projection $p$), will hold with probability 1 under any reasonable parameterization.

---

[4]"Almost all functions" in this context means, that for any continuous distribution over the parameters of the HT-model, with probability one the following statement is true for a function realized by an HT-model with sampled parameters.

We now state and prove a lemma that will be needed for our proof of theorem 1.

**Lemma 1.** *Let $M, N, K \in \mathbb{N}$, $1 \leq r \leq \min\{M, N\}$ and a polynomial mapping $A : \mathbb{R}^K \to \mathbb{R}^{M \times N}$ (i.e. for every $i \in [M], j \in [N]$ then $A_{ij} : \mathbb{R}^k \to \mathbb{R}$ is a polynomial function). If there exists a point $\mathbf{x} \in \mathbb{R}^K$ s.t. $\operatorname{rank}(A(\mathbf{x})) \geq r$, then the set $\{\mathbf{x} \in \mathbb{R}^K | \operatorname{rank}(A(\mathbf{x})) < r\}$ has zero measure.*

*Proof.* Remember that $\operatorname{rank}(A(\mathbf{x})) \geq r$ iff there exits a non-zero $r \times r$ minor of $A(\mathbf{x})$, which is polynomial in the entries of $A(\mathbf{x})$, and so it is polynomial in $\mathbf{x}$ as well. Let $c = \binom{M}{r} \cdot \binom{N}{r}$ be the number of minors in $A$, denote the minors by $\{f_i(\mathbf{x})\}_{i=1}^c$, and define the polynomial function $f(\mathbf{x}) = \sum_{i=1}^c f_i(\mathbf{x})^2$. It thus holds that $f(\mathbf{x}) = 0$ iff for all $i \in [c]$ it holds that $f_i(\mathbf{x}) = 0$, i.e. $f(\mathbf{x}) = 0$ iff $\operatorname{rank}(A(\mathbf{x})) < r$.

Now, $f(\mathbf{x})$ is a polynomial in the entries of $\mathbf{x}$, and so it either vanishes on a set of zero measure, or it is the zero polynomial (see Caron and Traynor (2005) for proof). Since we assumed that there exists $\mathbf{x} \in \mathbb{R}^K$ s.t. $\operatorname{rank}(A(\mathbf{x})) \geq r$, the latter option is not possible. $\qquad\square$

Following the work of Cohen et al. (2016a), our main proof relies on following notations and facts:

- We denote by $[\mathcal{A}]$ the matricization of an $N$-order tensor $\mathcal{A}$ (for simplicity, $N$ is assumed to be even), where rows and columns correspond to odd and even modes, respectively. Specifically, if $\mathcal{A} \in \mathbb{R}^{M_1 \times \cdots M_N}$, the matrix $[\mathcal{A}]$ has $M_1 \cdot M_3 \cdot \ldots \cdot M_{N-1}$ rows and $M_2 \cdot M_4 \cdot \ldots \cdot M_N$ columns, rearranging the entries of the tensor such that $\mathcal{A}_{d_1,\ldots,d_N}$ is stored in row index $1 + \sum_{i=1}^{N/2}(d_{2i-1} - 1)\prod_{j=i+1}^{N/2} M_{2j-1}$ and column index $1 + \sum_{i=1}^{N/2}(d_{2i} - 1)\prod_{j=i+1}^{N/2} M_{2j}$. Additionally, the matricization is a linear operator, i.e. for all scalars $\alpha_1, \alpha_2$ and tensors $\mathcal{A}_1, \mathcal{A}_2$ with the order and dimensions in every mode, it holds that $[\alpha_1 \mathcal{A}_1 + \alpha_2 \mathcal{A}_2] = \alpha_1[\mathcal{A}_1] + \alpha_2[\mathcal{A}_2]$.

- The relation between the Kronecker product (denoted by $\odot$) and the tensor product (denoted by $\otimes$) is given by $[\mathcal{A} \otimes \mathcal{B}] = [\mathcal{A}] \odot [\mathcal{B}]$.

- For any two matrices $A$ and $B$, it holds that $\operatorname{rank}(A \odot B) = \operatorname{rank}(A) \cdot \operatorname{rank}(B)$.

- Let $Z$ be the CP-rank of $\mathcal{A}$, then it holds that $\operatorname{rank}([\mathcal{A}]) \leq Z$ (see (Cohen et al., 2016a) for proof).

*Proof of theorem 1.* Stemming from the above stated facts, to show that the CP-rank of $\mathcal{A}^y$ is at least $r^{N/2}$, it is sufficient to examine its matricization $[\mathcal{A}^y]$ and prove that $\operatorname{rank}([\mathcal{A}^y]) \geq r^{N/2}$.

Notice from the construction of $[\mathcal{A}^y]$, according to the recursive formula of the HT-decomposition, that its entires are polynomial in the parameters of the decomposition, its dimensions are $M^{N/2}$ each and that $1 \leq r^{N/2} \leq M^{N/2}$. In accordance with the discussion on the measure of simplex spaces, for each vector parameter $\mathbf{a}^{l,j,\gamma} \in \triangle^{r_l-1}$, we instead examine its projection $\tilde{\mathbf{a}}^{l,j,\gamma} = p(\mathbf{a}^{l,j,\gamma}) \in \mathbb{R}^{r_l-1}$, and notice that $p^{-1}(\tilde{\mathbf{a}}^{l,j,\gamma})$ is a polynomial mapping[5] w.r.t. $\tilde{\mathbf{a}}^{l,j,\gamma}$. Thus, $[\mathcal{A}^y]$ is a polynomial mapping w.r.t. the projected parameters $\{\tilde{\mathbf{a}}^{l,j,\gamma}\}_{l,j,\gamma}$, and using lemma 1 it is sufficient to show that there exists a set of parameters for which $\operatorname{rank}([\mathcal{A}^y]) \geq r^{N/2}$.

Denoting for convenience $\phi^{L,1,1} := \mathcal{A}^y$ and $r_L = 1$, we will construct by induction over $l = 1, ..., L$ a set of parameters, $\{\mathbf{a}^{l,j,\gamma}\}_{l,j,\gamma}$, for which the ranks of the matrices $\{[\phi^{l,j,\gamma}]\}_{j\in[N/2^l],\gamma\in[r_l]}$ are at least $r^{2^l/2}$, while enforcing the simplex constraints on the parameters. More so, we'll construct these parameters s.t. $\mathbf{a}^{l,j,\gamma} = \mathbf{a}^{l,\gamma}$, thus proving both the "unshared" and "shared" cases.

For the case $l = 1$ we have:

$$\phi^{1,j,\gamma} = \sum_{\alpha=1}^{r_0} a_\alpha^{1,j,\gamma} \mathbf{a}^{0,2j-1,\alpha} \otimes \mathbf{a}^{0,2j,\alpha}$$

and let $a_\alpha^{1,j,\gamma} = \frac{1_{\alpha \leq r}}{r}$ and $a_i^{0,j,\alpha} = 1_{\alpha=i}$ for all $i, j, \gamma$ and $\alpha \leq M$, and $a_i^{0,j,\alpha} = 1_{i=1}$ for all $i$ and $\alpha > M$, and so

$$[\phi^{1,j,\gamma}]_{i,j} = \begin{cases} 1/r & i = j \wedge i \leq r \\ 0 & Otherwise \end{cases}$$

which means $\operatorname{rank}([\phi^{1,j,\gamma}]) = r$, while preserving the simplex constraints, which proves our inductive hypothesis for $l = 1$.

---

[5]As we mentioned earlier, $p$ is invertible only over $p(\triangle^k)$, for which its inverse is given by $p^{-1}(x_1, \ldots, x_k) = (x_1, \ldots, x_k, 1 - \sum_{i=1}^k x_i)$. However, to simplified the proof and notations, we use $p^{-1}$ as defined here over the entire range $\mathbb{R}^{k-1}$, even where it does not serve as the inverse of $p$.

Assume now that rank $\left([\phi^{l-1,j',\gamma'}]\right) \geq r^{2^{l-1}/2}$ for all $j' \in [N/2^{l-1}]$ and $\gamma' \in [r_{l-1}]$. For some specific choice of $j \in [N/2^l]$ and $\gamma \in [r_l]$ we have:

$$\phi^{l,j,\gamma} = \sum_{\alpha=1}^{r_{l-1}} a_\alpha^{l,j,\gamma} \phi^{l-1,2j-1,\alpha} \otimes \phi^{l-1,2j,\alpha}$$

$$\implies [\phi^{l,j,\gamma}] = \sum_{\alpha=1}^{r_{l-1}} a_\alpha^{l,j,\gamma} [\phi^{l-1,2j-1,\alpha}] \odot [\phi^{l-1,2j,\alpha}]$$

Denote $M_\alpha := [\phi^{l-1,2j-1,\alpha}] \odot [\phi^{l-1,2j,\alpha}]$ for $\alpha = 1,...,r_{l-1}$. By our inductive assumption, and by the general property rank $(A \odot B) = $ rank $(A) \cdot$ rank $(B)$, we have that the ranks of all matrices $M_\alpha$ are at least $r^{2^{l-1}/2} \cdot r^{2^{l-1}/2} = r^{2^l/2}$. Writing $[\phi^{l,j,\gamma}] = \sum_{\alpha=1}^{r_{l-1}} a_\alpha^{l,j,\gamma} \cdot M_\alpha$, and noticing that $\{M_\alpha\}$ do not depend on $\mathbf{a}^{l,j,\gamma}$, we simply pick $a_\alpha^{l,j,\gamma} = 1_{\alpha=1}$, and thus $\phi^{l,j,\gamma} = M_1$, which is of rank $r^{2^l/2}$. This completes the proof of the theorem. □

From the perspective of TMMs, theorem 1 leads to the following corollary:

**Corollary 3.** *Assume the mixing components $\mathcal{M} = \{f_i(\mathbf{x}) \in L^2(\mathbb{R}^2) \cap L^1(\mathbb{R}^s)\}_{i=1}^M$ are square integrable[6] probability density functions, which form a linearly independent set. Consider a deep GHT-model of polynomial size whose parameters are drawn at random by some continuous distribution. Then, with probability 1, the distribution realized by this network requires an exponential size in order to be realized (or approximated w.r.t. the $L^2$ distance) by the shallow GCP-model. The claim holds regardless of whether the parameters of the deep GHT-model are shared or not.*

*Proof.* Given a coefficient tensor $\mathcal{A}$, the CP-rank of $\mathcal{A}$ is a lower bound on the number of channels (denoted by $Z$ in the body of the article) required to represent that tensor by the ConvAC following the CP decomposition as introduced in sec. 2. Additionally, since the mixing components are linearly independent, their products $\{\prod_{i=1}^N f_i(\mathbf{x}_i) | f_i \in \mathcal{M}\}$ are linearly independent as well, which entails that any distribution representable by the TMM with mixing components $\mathcal{M}$ has a unique coefficient tensor $\mathcal{A}$. From theorem 1, the set of parameters of a polynomial GHT-model with a coefficient tensor of a polynomial CP-rank, the requirement for a polynomial GCP-model realizing that distribution exactly, forms a set of measure zero.

It is left to prove, that not only is it impossible to exactly represent a distribution with an exponential coefficient tensor by a GCP-model, it is also impossible to approximate it. This follows directly from lemma 7 in appendix B of Cohen et al. (2016a), as our case meets the requirement of that lemma. □

# E    PROOF FOR THE OPTIMALITY OF MARGINALIZED BAYES PREDICTOR

In this section we give short proofs for the claims from sec. 5, on the optimality of the marginalized Bayes predictor under missing-at-random (MAR) distribution, when the missingness mechanism is unknown, as well as the general case when we do not add additional assumptions. In addition, we will also present a counter example proving data imputation results lead to suboptimal classification performance. We begin by introducing several notations that augment the notations already introduced in the body of the article.

Given a specific mask realization $\mathbf{m} \in \{0,1\}^s$, we use the following notations to denote partial assignments to the random vector $\mathcal{X}$. For the observed indices of $\mathcal{X}$, i.e. the indices for which $m_i = 1$, we denote a partial assignment by $\mathcal{X} \setminus \mathbf{m} = \mathbf{x}_o$, where $\mathbf{x}_o \in \mathbb{R}^{d_o}$ is a vector of length $d_o$ equal to the number of observed indices. Similarly, we denote by $\mathcal{X} \cap \mathbf{m} = \mathbf{x}_m$ a partial assignment to the missing indices according to $\mathbf{m}$, where $\mathbf{x}_m \in \mathbb{R}^{d_m}$ is a vector of length $d_m$ equal to the number of missing indices. As an example of the notation, for given realizations $\mathbf{x} \in \mathbb{R}^s$ and $\mathbf{m} \in \{0,1\}^s$, we defined in sec. 5 the event $o(\mathbf{x}, \mathbf{m})$, which using current notation is marked by the partial assignment $\mathcal{X} \setminus \mathbf{m} = \mathbf{x}_o$ where $\mathbf{x}_o$ matches the observed values of the vector $\mathbf{x}$ according to $\mathbf{m}$.

With the above notations in place, we move on to prove claim 1, which describes the general solution to the optimal prediction rule given both the data and missingness distributions, and without adding any additional assumptions.

---

[6]It is important to note that most commonly used distribution functions are square integrable, e.g. most members of the exponential family such as the Gaussian distribution.

*Proof of claim 1.* Fix an arbitrary prediction rule $h$. We will show that $L(h^*) \leq L(h)$, where $L$ is the expected 0-1 loss.

$$1 - L(h) = E_{(\mathbf{x},\mathbf{m},y) \sim (\mathcal{X},\mathcal{M},\mathcal{Y})} [1_{h(\mathbf{x} \odot \mathbf{m}) = y}]$$

$$= \sum_{\mathbf{m} \in \{0,1\}^s} \sum_{y \in [k]} \int_{\mathbb{R}^s} \mathcal{P}(\mathcal{M}=\mathbf{m}, \mathcal{X}=\mathbf{x}, \mathcal{Y}=y) 1_{h(\mathbf{x} \odot \mathbf{m}) = y} d\mathbf{x}$$

$$= \sum_{\mathbf{m} \in \{0,1\}^s} \sum_{y \in [k]} \int_{\mathbb{R}^{d_o}} \int_{\mathbb{R}^{d_m}} \mathcal{P}(\mathcal{M}=\mathbf{m}, \mathcal{X} \backslash \mathbf{m} = \mathbf{x}_o, \mathcal{X} \cap \mathbf{m} = \mathbf{x}_m, \mathcal{Y}=y) 1_{h(\mathbf{x} \otimes \mathbf{m}) = y} d\mathbf{x}_o d\mathbf{x}_m$$

$$=_1 \sum_{\mathbf{m} \in \{0,1\}^s} \sum_{y \in [k]} \int_{\mathbb{R}^{d_o}} 1_{h(\mathbf{x} \odot \mathbf{m}) = y} d\mathbf{x}_o \int_{\mathbb{R}^{d_m}} \mathcal{P}(\mathcal{M}=\mathbf{m}, \mathcal{X} \backslash \mathbf{m} = \mathbf{x}_o, \mathcal{X} \cap \mathbf{m} = \mathbf{x}_m, \mathcal{Y}=y) d\mathbf{x}_m$$

$$=_2 \sum_{\mathbf{m} \in \{0,1\}^s} \sum_{y \in [k]} \int_{\mathbb{R}^{d_o}} 1_{h(\mathbf{x} \odot \mathbf{m}) = y} \mathcal{P}(\mathcal{M}=\mathbf{m}, \mathcal{X} \backslash \mathbf{m} = \mathbf{x}_o, \mathcal{Y}=y) d\mathbf{x}_o$$

$$=_3 \sum_{\mathbf{m} \in \{0,1\}^s} \int_{\mathbb{R}^{d_o}} \mathcal{P}(\mathcal{X} \backslash \mathbf{m} = \mathbf{x}_o) \sum_{y \in [k]} 1_{h(\mathbf{x} \odot \mathbf{m}) = y} \mathcal{P}(\mathcal{Y}=y | \mathcal{X} \backslash \mathbf{m} = \mathbf{x}_o) \mathcal{P}(\mathcal{M}=\mathbf{m} | \mathcal{X} \backslash \mathbf{m} = \mathbf{x}_o, \mathcal{Y}=y) d\mathbf{x}_o$$

$$\leq_4 \sum_{\mathbf{m} \in \{0,1\}^s} \int_{\mathbb{R}^{d_o}} \mathcal{P}(\mathcal{X} \backslash \mathbf{m} = \mathbf{x}_o) \sum_{y \in [k]} 1_{h^*(\mathbf{x} \odot \mathbf{m}) = y} \mathcal{P}(\mathcal{Y}=y | \mathcal{X} \backslash \mathbf{m} = \mathbf{x}_o) \mathcal{P}(\mathcal{M}=\mathbf{m} | \mathcal{X} \backslash \mathbf{m} = \mathbf{x}_o, \mathcal{Y}=y) d\mathbf{x}_o$$

$$= 1 - L(h^*)$$

Where (1) is because the output of $h(\mathbf{x} \odot \mathbf{m})$ is independent of the missing values, (2) by marginalization, (3) by conditional probability definition and (4) because by definition $h^*(\mathbf{x} \odot \mathbf{m})$ maximizes the expression $\mathcal{P}(\mathcal{Y}=y | \mathcal{X} \backslash \mathbf{m} = \mathbf{x}_o) \mathcal{P}(\mathcal{M}=\mathbf{m} | \mathcal{X} \backslash \mathbf{m} = \mathbf{x}_o, \mathcal{Y}=y)$ w.r.t. the possible values of $y$ for fixed vectors $\mathbf{m}$ and $\mathbf{x}_o$. Finally, by replacing integrals with sums, the proof holds exactly the same when instances ($\mathcal{X}$) are discrete. □

We now continue and prove corollary 1, a direct implication of claim 1 which shows that in the MAR setting, the missingness distribution can be ignored, and the optimal prediction rule is given by the marginalized Bayes predictor.

*Proof of corollary 1.* Using the same notation as in the previous proof, and denoting by $\mathbf{x}_o$ the partial vector containing the observed values of $\mathbf{x} \odot \mathbf{m}$, the following holds:

$$\mathcal{P}(\mathcal{M}=\mathbf{m} | o(\mathbf{x}, \mathbf{m}), \mathcal{Y}=y) := \mathcal{P}(\mathcal{M}=\mathbf{m} | \mathcal{X} \backslash \mathbf{m} = \mathbf{x}_o, \mathcal{Y}=y)$$

$$= \int_{\mathbb{R}^{d_m}} \mathcal{P}(\mathcal{M}=\mathbf{m}, \mathcal{X} \cap \mathbf{m} = \mathbf{x}_m | \mathcal{X} \backslash \mathbf{m} = \mathbf{x}_o, \mathcal{Y}=y) d\mathbf{x}_m$$

$$= \int_{\mathbb{R}^{d_m}} \mathcal{P}(\mathcal{X} \cap \mathbf{m} = \mathbf{x}_m | \mathcal{X} \backslash \mathbf{m} = \mathbf{x}_o, \mathcal{Y}=y) \cdot \mathcal{P}(\mathcal{M}=\mathbf{m} | \mathcal{X} \cap \mathbf{m} = \mathbf{x}_m, \mathcal{X} \backslash \mathbf{m} = \mathbf{x}_o, \mathcal{Y}=y) d\mathbf{x}_m$$

$$=_1 \int_{\mathbb{R}^{d_m}} \mathcal{P}(\mathcal{X} \cap \mathbf{m} = \mathbf{x}_m | \mathcal{X} \backslash \mathbf{m} = \mathbf{x}_o, \mathcal{Y}=y) \cdot \mathcal{P}(\mathcal{M}=\mathbf{m} | \mathcal{X} \cap \mathbf{m} = \mathbf{x}_m, \mathcal{X} \backslash \mathbf{m} = \mathbf{x}_o) d\mathbf{x}_m$$

$$=_2 \int_{\mathbb{R}^{d_m}} \mathcal{P}(\mathcal{X} \cap \mathbf{m} = \mathbf{x}_m | \mathcal{X} \backslash \mathbf{m} = \mathbf{x}_o, \mathcal{Y}=y) \cdot \mathcal{P}(\mathcal{M}=\mathbf{m} | \mathcal{X} \backslash \mathbf{m} = \mathbf{x}_o) d\mathbf{x}_m$$

$$= \mathcal{P}(\mathcal{M}=\mathbf{m} | \mathcal{X} \backslash \mathbf{m} = \mathbf{x}_o) \int_{\mathbb{R}^{d_m}} \mathcal{P}(\mathcal{X} \cap \mathbf{m} = \mathbf{x}_m | \mathcal{X} \backslash \mathbf{m} = \mathbf{x}_o, \mathcal{Y}=y) d\mathbf{x}_m$$

$$= \mathcal{P}(\mathcal{M}=\mathbf{m} | o(\mathbf{x}, \mathbf{m}))$$

Where (1) is due to the independence assumption of the events $\mathcal{Y} = y$ and $\mathcal{M} = \mathbf{m}$ conditioned on $\mathcal{X} = \mathbf{x}$, while noting that $(\mathcal{X} \backslash \mathbf{m} = x_o) \wedge (\mathcal{X} \cap \mathbf{m} = x_m)$ is a complete assignment of $\mathcal{X}$. (2) is due to the MAR assumption, i.e. that for a given $\mathbf{m}$ and $\mathbf{x}_o$ it holds for all $\mathbf{x}_m \in \mathbb{R}^{d_m}$:

$$\mathcal{P}(\mathcal{M}=\mathbf{m} | \mathcal{X} \backslash \mathbf{m} = \mathbf{x}_o, \mathcal{X} \cap \mathbf{m} = \mathbf{x}_m) = \mathcal{P}(\mathcal{M}=\mathbf{m} | \mathcal{X} \backslash \mathbf{m} = \mathbf{x}_o)$$

We have shown that $\mathcal{P}(\mathcal{M}=\mathbf{m} | o(\mathbf{x}, \mathbf{m}), \mathcal{Y} = y)$ does not depend on $y$, and thus does not affect the optimal prediction rule in claim 1. It may therefore be dropped, and we obtain the marginalized Bayes predictor. □

Having proved that in the MAR setting, classification through marginalization leads to optimal performance, we now move on to show that the same is not true for classification through data-imputation. Though there are many methods to perform data-imputation, i.e. to complete missing values given the observed ones, all of these methods can be seen as the solution of the following optimization problem, or more typically its approximation:

$$g(\mathbf{x} \odot \mathbf{m}) = \operatorname*{argmax}_{\mathbf{x}' \in \mathbb{R}^s \wedge \forall i: m_i = 1 \rightarrow x'_i = x_i} \mathcal{P}(\mathcal{X} = \mathbf{x}')$$

| $X_1$ | $X_2$ | $Y$ | Weight | Probability ($\epsilon = 10^{-4}$) |
|---|---|---|---|---|
| 0 | 0 | 0 | $1 - \epsilon$ | 16.665% |
| 0 | 1 | 0 | 1 | 16.667% |
| 1 | 0 | 0 | $1 - \epsilon$ | 16.665% |
| 1 | 1 | 0 | 1 | 16.667% |
| 0 | 0 | 1 | 0 | 0.000% |
| 0 | 1 | 1 | $1 + \epsilon$ | 16.668% |
| 1 | 0 | 1 | 0 | 0.000% |
| 1 | 1 | 1 | $1 + \epsilon$ | 16.668% |

Table 2: Data distribution over the space $\mathcal{X} \times \mathcal{Y} = \{0,1\}^2 \times \{0,1\}$ that serves as the example for the sub-optimality of classification through data-imputation (proof of claim 3).

Where $g(\mathbf{x} \odot \mathbf{m})$ is the most likely completion of $\mathbf{x} \odot \mathbf{m}$. When data-imputation is carried out for classification purposes, one is often interested in data-imputation conditioned on a given class $Y = y$, i.e.:

$$g(\mathbf{x} \odot \mathbf{m}; y) = \underset{\mathbf{x}' \in \mathbb{R}^s \wedge \forall i : m_i = 1 \rightarrow x_i' = x_i}{\operatorname{argmax}} \mathcal{P}(\mathcal{X} = \mathbf{x}' | \mathcal{Y} = y)$$

Given a classifier $h : \mathbb{R}^s \rightarrow [K]$ and an instance $\mathbf{x}$ with missing values according to $\mathbf{m}$, classification through data-imputation is simply the result of applying $h$ on the output of $g$. When $h$ is the optimal classifier for complete data, i.e. the Bayes predictor, we end up with one of the following prediction rules:

$$\text{Unconditional:} \quad h(\mathbf{x} \odot \mathbf{m}) = \underset{y}{\operatorname{argmax}} \, \mathcal{P}(\mathcal{Y} = y | \mathcal{X} = g(\mathbf{x} \odot \mathbf{m}))$$

$$\text{Conditional:} \quad h(\mathbf{x} \odot \mathbf{m}) = \underset{y}{\operatorname{argmax}} \, \mathcal{P}(\mathcal{Y} = y | \mathcal{X} = g(\mathbf{x} \odot \mathbf{m}; y))$$

**Claim 3.** *There exists a data distribution $\mathcal{D}$ and MAR missingness distribution $\mathcal{Q}$ s.t. the accuracy of classification through data-imputation is almost half the accuracy of the optimal marginalized Bayes predictor, with an absolute gap of more than 33 percentage points.*

*Proof.* For simplicity, we will give an example for a discrete distribution over the binary set $\mathcal{X} \times \mathcal{Y} = \{0,1\}^2 \times \{0,1\}$. Let $1 > \epsilon > 0$ be some small positive number, and we define $\mathcal{D}$ according to table 2, where each triplet $(x_1, x_2, y) \in \mathcal{X} \times \mathcal{Y}$ is assigned a positive weight, which through normalization defines a distribution over $\mathcal{X} \times \mathcal{Y}$. The missingness distribution $\mathcal{Q}$ is defined s.t. $P_{\mathcal{Q}}(M_1 = 1, M_2 = 0 | X = \mathbf{x}) = 1$ for all $\mathbf{x} \in \mathcal{X}$, i.e. $X_1$ is always observed and $X_2$ is always missing, which is a trivial MAR distribution. Given the above data distribution $\mathcal{D}$, we can easily calculate the exact accuracy of the optimal data-imputation classifier and the marginalized Bayes predictor under the missingness distribution $\mathcal{Q}$, as well as the standard Bayes predictor under full-observability. First notice that whether we apply conditional or unconditional data-imputation, and whether $X_1$ is equal to 0 or 1, the completion will always be $X_2 = 1$ and the predicted class will always be $Y = 1$. Since the data-imputation classifiers always predict the same class $Y = 1$ regardless of their input, the probability of success is simply the probability $P(Y = 1) = \frac{1+\epsilon}{3}$ (for $\epsilon = 10^{-4}$ it equals approximately 33.337%). Similarly, the marginalized Bayes predictor always predicts $Y = 0$ regardless of its input, and so its probability of success is $P(Y = 0) = \frac{2-\epsilon}{3}$ (for $\epsilon = 10^{-4}$ it equals approximately 66.663%), which is almost double the accuracy achieved by the data-imputation classifier. Additionally, notice that the marginalized Bayes predictor achieves almost the same accuracy as the Bayes predictor under full-observability, which equals exactly $\frac{2}{3}$. $\quad\square$

## F  DETAILED DESCRIPTION OF THE EXPERIMENTS

Experiments are meaningful only if they could be reproduced by other proficient individuals. Providing sufficient details to enable others to replicate our results is the goal of this section. We hope to accomplish this by making our code public, as well as documenting our experiments to a sufficient degree allowing for their reproduction from scratch. Our complete implementation of the models presented in this paper, as well as our modifications to other open-source projects and scripts used in the process of conducting our experiments, are available at our Github repository: `https://github.com/HUJI-Deep/TMM`. We additionally wish to invite readers to contact the authors, if they deem the following details insufficient in their process to reproduce our results.

### F.1  DESCRIPTION OF METHODS

In the following we give concise descriptions of each classification method we have used in our experiments. The results of the experiment on MP-DBM (Goodfellow et al., 2013) were taken directly from the paper and

were not conducted by us, hence we do not cover it in this section. We direct the reader to that article for exact details on how to reproduce their results.

### F.1.1 ROBUST LINEAR CLASSIFIER

In Dekel and Shamir (2008), binary linear classifiers were trained by formulating their optimization as a quadric program under the constraint that some of its features could be deleted, i.e. their original value was changed to zero. While the original source code was never published, the authors have kindly agreed to share with us their code, which we used to reproduced their results, but on larger datasets. The algorithm has only a couple hyper-parameters, which were chosen by a grid-search through a cross-validation process. For details on the exact protocol for testing binary classifiers on missing data, please see sec. F.2.1.

### F.1.2 K-NEAREST NEIGHBORS

K-Nearest Neighbors (KNN) is a classical machine learning algorithm used for both regression and classification tasks. Its underlying mechanism is finding the $k$ nearest examples (called neighbors) from the training set, $(\mathbf{x}_1, y_1), \ldots, (\mathbf{x}_k, y_k) \in S$, according to some metric function $d(\cdot, \cdot) : \mathcal{X} \times \mathcal{X} \rightarrow \mathbb{R}_+$, after which a summarizing function $f$ is applied to the targets of the $k$ nearest neighbors to produce the output $y^* = f(y_1, \ldots, y_k)$. When KNN is used for classification, $f$ is typically the majority voting function, returning the class found in most of the $k$ nearest neighbors.

In our experiments we use KNN for classification with missing data, where the training set consists of complete examples with no missing data, but at classification time the inputs have missing values. Given an input with missing values $\mathbf{x} \odot \mathbf{m}$ and an example $\mathbf{x}'$ from the training set, we use a modified Euclidean distance metric, where we compare the distance only against the non-missing coordinates of $\mathbf{x}$, i.e. the metric is defined by $d(\mathbf{x}', \mathbf{x} \odot \mathbf{m}) = \sum_{i:m_i=1} (x_i' - x_i)^2$. Through a process of cross-validation we have chosen $k = 5$ for all of our experiments. Our implementation of KNN is based on the popular *scikit-learn* python library (Pedregosa et al., 2011).

### F.1.3 CONVOLUTIONAL NEURAL NETWORKS

The most widespread and successful discriminative method nowadays are Convolutional Neural Networks (ConvNets). Standard ConvNets are represented by a computational graph consisted of different kinds of nodes, called layers, with a convolutional-like operators applied to their inputs, followed by a non-linear point-wise activation function, e.g. $\max(0, x)$ known as ReLU.

For our experiments on MNIST, both with and without missing data, we have used the LeNeT ConvNet architecture (LeCun et al., 1998) that is bundled with Caffe (Jia et al., 2014), trained for 20,000 iterations using SGD with 0.9 momentum and 0.01 base learning rate, which remained constant for 10,000 iterations, followed by a linear decrease to 0.001 for another 5,000 iterations, followed by a linear decrease to 0 learning rate for the remaining 5,000 iterations. The model also used $l_2$-regularization (also known as weight decay), which was chosen through cross-validation for each experiment separately. No other modifications were made to the model or its training procedure.

For our experiments on NORB, we have used an ensemble of 3 ConvNets, each using the following architecture: $5 \times 5$ convolution with 128 output channels, $3 \times 3$ max pooling with stride 2, ReLU activation, $5 \times 5$ convolution with 128 output channels, ReLU activation, dropout layer with probability 0.5, $3 \times 3$ average pooling with stride 2, $5 \times 5$ convolution with 256 output channels, ReLU activation, dropout layer with probability 0.5, $3 \times 3$ average pooling with stride 2, fully-connected layer with 768 output channels, ReLU activation, dropout layer with probability 0.5, and ends with fully-connected layer with 5 output channels. The stereo images were represented as a two-channel input image when fed to the network. During training we have used data augmentation consisting of randomly scaling and rotation transforms. The networks were trained for 40,000 iterations using SGD with 0.99 momentum and 0.001 base learning rate, which remained constant for 30,000 iterations, followed by a linear decrease to 0.0001 for 6000 iterations, followed by a linear decrease to 0 learning rate for the remaining 4,000 iterations. The model also used 0.0001 weight decay for additional regularization.

When ConvNets were trained on images containing missing values, we passed the network the original image with missing values zeroed out, and an additional binary image as a separate channel, containing 1 for missing values at the same spatial position, and 0 otherwise – this missing data format is sometimes known as *flag data imputation*. Other formats for representing missing values were tested (e.g. just using zeros for missing values), however, the above scheme performed significantly better than other formats. In our experiments, we assumed that the training set was complete and missing values were only present in the test set. In order to design ConvNets that are robust against specific missingness distributions, we have simulated missing values during training, sampling a different mask of missing values for each image in each mini-batch. As covered in sec. 6, the results of training ConvNets directly on simulated missingness distributions resulted in classifiers

which were biased towards the specific distribution used in training, and performed worse on other distributions compared to ConvNets trained on the same distribution.

In addition to training ConvNets directly on missing data, we have also used them as the classifier for testing different data imputation methods, as describe in the next section.

### F.1.4 CLASSIFICATION THROUGH DATA IMPUTATION

The most common method for handling missing data, while leveraging available discriminative classifiers, is through the application of *data imputation* – an algorithm for the completion of missing values – and then passing the results to a classifier trained on uncorrupted dataset. We have tested five different types of data imputation algorithms:

- Zero data imputation: replacing every missing value by zero.

- Mean data imputation: replacing every missing value by the mean value computed over the dataset.

- Generative data imputation: training a generative model and using it to complete the missing values by finding the most likely instance that coincides with the observed values, i.e. solving the following

$$g(\mathbf{x} \odot \mathbf{m}) = \underset{\mathbf{x}' \in \mathbb{R}^s \wedge \forall i, m_i = 1 \rightarrow x'_i = x_i}{\operatorname{argmax}} P(X = \mathbf{x}')$$

  We have tested the following generative models:

  - Generative Stochastic Networks (GSN) (Bengio et al., 2014): We have used their original source code from `https://github.com/yaoli/GSN`, and trained their example model on MNIST for 1000 epochs. Whereas in the original article they have tested completing only the left or right side of a given image, we have modified their code to support general masks. Our modified implementation can be found at `https://github.com/HUJI-Deep/GSN`.
  - Non-linear Independent Components Estimation (NICE) (Dinh et al., 2014): We have used their original source code from `https://github.com/laurent-dinh/nice`, and trained it on MNIST using their example code without changes. Similarly to our modification to the GSN code, here too we have adapted their code to support general masks over the input. Additionally, their original inpainting code required 110,000 iterations, which we have reduced to just 8,000 iterations, since the effect on classification accuracy was marginal. For the NORB dataset, we have used their CIFAR10 example, with lower learning rate of $10^{-4}$. Our modified code can be found at `https://github.com/HUJI-Deep/nice`.
  - Diffusion Probabilistic Models (DPM) (Sohl-Dickstein et al., 2015): We have user their original source code from `https://github.com/Sohl-Dickstein/Diffusion-Probabilistic-Models`, and trained it on MNIST using their example code without changes. Similarly to our modifications to GSN, we have add support for a general mask of missing values, but other than that kept the rest of the parameters for inpainting unchanged. For NORB we have used the same model as MNIST. We have tried using their CIFAR10 example for NORB, however, it produced exceptions during training. Our modified code can be found at `https://github.com/HUJI-Deep/Diffusion-Probabilistic-Models`.

### F.1.5 TENSORIAL MIXTURE MODELS

For a complete theoretical description of our model please see the body of the article. Our models were implemented by performing all intermediate computations in log-space, using numerically aware operations. In practiced, that meant our models were realized by the SimNets architecture (Cohen and Shashua, 2014; Cohen et al., 2016b), which consists of Similarity layers representing gaussian distributions, MEX layers representing weighted sums performed on log-space input and outputs, as well as standard pooling operations. The learned parameters of the MEX layers are called *offsets*, which represents the weights of the weighted sum, but saved in log-space. The parameters of the MEX layers can be optionally shared between spatial regions, or alternatively left with no parameter sharing at all. Additionally, when used to implement our generative models, the offsets are normalized to have a soft-max (i.e., $\log\left(\sum_i \exp(x_i)\right)$) of zero.

The network architectures we have tested in this article, consists of $M$ different Gaussian mixture components with diagonal covariance matrices, over non-overlapping patches of the input of size $2 \times 2$, which were implemented by a similarity layer as specified by the SimNets architecture, but with an added gaussian normalization term.

We first describe the architectures used for the MNIST dataset. For the GCP-model, we used $M = 800$, and following the similarity layer is a $1 \times 1$ MEX layer with no parameter sharing over spatial regions and 10 output channels. The model ends with a global sum pooling operation, followed by another $1 \times 1$ MEX layer

with 10 outputs, one for each class. The GHT-model starts with the similarity layer with $M = 32$, followed by a sequence of four pairs of $1 \times 1$ MEX layer followed by $2 \times 2$ sum pooling layer, and after the pairs and additional $1 \times 1$ MEX layer lowering the outputs of the model to 10 outputs as the number of classes. The number of output channels for each MEX layer are as follows 64-128-256-512-10. All the MEX layers in this network do not use parameter sharing, except the first MEX layer, which uses a repeated sharing pattern of $2 \times 2$ offsets, that analogous to a $2 \times 2$ convolution layer with stride 2. Both models were trained with the losses described in sec. 4, using the Adam SGD variant for optimizing the parameters, with a base learning rate of 0.03, and $\beta_1 = \beta_2 = 0.9$. The models were trained for 25,000 iterations, where the learning rate was dropped by 0.1 after 20,000 iterations.

For the NORB dataset, we have trained only the GHT-model with $M = 128$ for the similarity layer. The MEX layers use the same parameter sharing scheme as the one for MNIST, and the number of output channels for each MEX layer are as follows: 256-256-256-512-5. Training was identical to the MNIST models, with the exception of using 40,000 iterations instead of just 25,000. Additionally, we have used an ensemble of 4 models trained separately, each trained using a different generative loss weight (see below for more information). We have also used the same data augmentation methods (scaling and rotation) which were used in training the ConvNets for NORB used in this article.

The standard $L_2$ weight regularization (sometimes known as weight decay) did not work well on our models, which lead us to adapt it to better fit to log-space weights, by minimizing $\lambda \sum_i \left( \exp\left( x_i \right) \right)^2$ instead of $\lambda ||\mathbf{x}||_2 = \lambda \sum_i \mathbf{x}_i^2$, where the parameter $\lambda$ was chosen through cross-validation. Additionally, since even with large values of $\lambda$ our model was still overfitting, we have added another form of regularization in the form of *random marginalization* layers. A random marginalization layer, is similar in concept to dropout, but instead of zeroing activations completely in random, it choses spatial locations at random, and then zero out the activations at those locations for all the channels. Under our model, zeroing all the activations in a layer at a specific location, is equivalent to marginalizing over all the inputs for the receptive field for that respective location. We have used random marginalization layers in between all our layers during training, where the probability for zeroing out activations was chosen through cross-validation for each layer separately. Though it might raise concern that random marginalization layers could lead to biased results toward the missingness distributions we have tested it on, in practice the addition of those layers only helped improve our results under cases where only few pixels where missing.

Finally, we wish to discuss a few optimization tricks which had a minor effects compared to the above, but were nevertheless very useful in achieving slightly better results. First, instead of optimizing directly the objective defined by eq. 4, we add smoothing parameter $\beta$ between the two terms, as follows:

$$\Theta^* = \underset{\Theta}{\mathrm{argmin}} - \sum_{i=1}^{|S|} \log \frac{e^{N_\Theta(X^{(i)}; Y^{(i)})}}{\sum_{y=1}^{K} e^{N_\Theta(X^{(i)}; y)}} - \beta \sum_{i=1}^{|S|} \log \sum_{y=1}^{K} e^{N_\Theta(X^{(i)}; y)}$$

setting $\beta$ too low diminish the generative capabilities of our models, while setting it too high diminish the discriminative performance. Through cross-validation, we decided on the value $\beta = 0.01$ for the models trained on MNIST, while for NORB we have used a different value of $\beta$ for each of the models, ranging in $\{0.01, 0.1, 0.5, 1\}$. Second, we found that performance increased if we normalized activations before applying the $1 \times 1$ MEX operations. Specifically, we calculate the soft-max over the channels for each spatial location which we call the activation norm, and then subtract it from every respective activation. After applying the MEX operation, we add back the activation norm. Though might not be obvious at first, subtracting a constant from the input of a MEX operation and adding it to its output is equivalent does not change the mathematical operation. However, it does resolve the numerical issue of adding very large activations to very small offsets, which might result in a loss of precision. Finally, we are applying our model in different translations of the input and then average the class predictions. Since our model can marginalize over inputs, we do not need to crop the original image, and instead mask the unknown parts after translation as missing. Applying a similar trick to standard ConvNets on MNIST does not seem to improve their results. We believe this method is especially fit to our model, is because it does not have a natural treatment of overlapping patches like ConvNets do, and because it is able to marginalize over missing pixels easily, not limiting it just to crop translation as is typically done.

## F.2 DESCRIPTION OF EXPERIMENTS

In this section we will give a detailed description of the protocol we have used during our experiments.

### F.2.1 BINARY DIGIT CLASSIFICATION WITH FEATURE DELETION MISSING DATA

This experiment focuses on the binary classification problem derived from MNIST, by limiting the number of classes to two different digits at a time. We use the same non-zero feature deletion distribution as suggested by Globerson and Roweis (2006), i.e. for a given image we uniformly sample a set of $N$ non-zero pixels from the

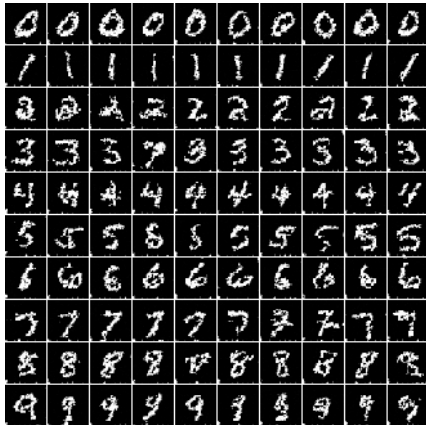

Figure 8: Generated digits samples from the GHT-model trained on the MNIST dataset.

image (if the image has less than $N$ non-zero pixels then they are non-zero pixels are chosen), and replace their values with zeros. This type of missingness distribution falls under the MNAR type defined in sec.5.

We test values of $N$ in $\{0, 25, 50, 75, 100, 125, 150\}$. For a given value of $N$, we train a separate classifier on each digit pair classifier on a randomly picked subset of the dataset containing 300 images per digit (600 total). During training we use a fixed validation set with 1000 images per digit. After picking the best classifier according to the validation set, the classifier is tested against a test set with a 1000 images per digits with a randomly chosen missing values according to the value of $N$. This experiment is repeated 10 times for each digit pair, each time using a different subset for the training set, and a new corrupted test set. After conducting all the different experiments, all the accuracies are averaged for each value of $N$, which are reported in table 1.

### F.2.2 Multi-class Digit Classification with MAR Missing Data

This experiment focuses on the complete multi-class digit classification of the MNIST dataset, in the presence of missing data according to different missingness distributions. Under this setting, only the test set contains missing values, whereas the training set does not. We test two kinds of missingness distributions, which both fall under the MAR type defined in sec.5. The first kind, which we call *i.i.d. corruption*, each pixel is missing with a fixed probability $p$. the second kind, which we call *missing rectangles corruption*, The positions of $N$ rectangles of width $W$ or chosen uniformly in the picture, where the rectangles can overlap one another. During the training stage, the models to be tested are not to be biased toward the specific missingness distributions we have chosen, and during the test stage, the same classifier is tested against all types of missingness distributions, and without supplying it with the parameters or type of the missingness distribution it is tested against. This rule prevent the use of ConvNets trained on simulated missingness distributions. To demonstrate that the latter lead to biased classifiers, we have conducted a separate experiment just for ConvNets, where the previous rule is ignored, and we train a separate ConvNet classifier on each type and parameter of the missingness distributions we have used. We then tested each of those ConvNets on all other missingness distributions, the results of which are in fig. 5, which confirmed our hypothesis.

## G  Image Generation and Network Visualization

Following the graphical model perspective of our models allows us to not only generate random instances from the distribution, but to also generate the most likely patches for each neuron in the network, effectively explaining its role in the classification process. We remind the reader that every neuron in the network corresponds to a possible assignment of a latent variable in the graphical model. By looking for the most likely assignments for each of its child nodes in the graphical tree model, we can generate a patch that describes that neuron. Unlike similar suggested methods to visualize neural networks (Zeiler and Fergus, 2014), often relying on brute-force search or on solving some optimization problem to find the most likely image, our method emerges naturally from the probabilistic interpretation of our model.

In fig. 8, we can see conditional samples generates for each digit, while in fig. 9 we can see a visualization of the top-level layers of network, where each small patch matches a different neuron in the network. The common wisdom of how ConvNets work is by assuming that simple low-level features are composed together to create more and more complex features, where each subsequent layer denotes features of higher abstraction – the visualization of our network clearly demonstrate this hypothesis to be true for our case, showing small strokes iteratively being composed into complete digits.

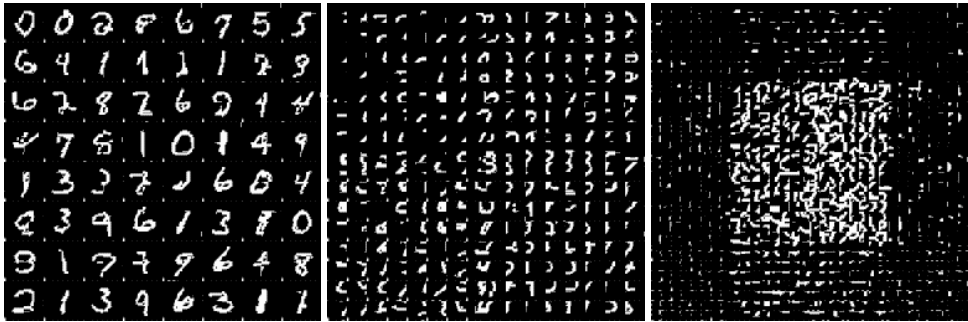

Figure 9: Visualization of the GHT-model. Each of the images above visualize a different layer of the model and consists of several samples generated from latent variables at different spatial locations conditioned on randomly selected channels. The leftmost image shows samples taken from the 5th layer which consists of just a single latent variable with 512 channels. The center image shows samples taken from the 4th layer, which consists of $2 \times 2$ grid of latent variables with 256 channels each. The image is divided to 4 quadrants, each contains samples taken from the respective latent variable at that position. The rightmost image shows samples from the 3rd layer, which consists of $4 \times 4$ grid of latent variables with 128 channels, and the image is similarly spatial divided into different areas matching the latent variables of the layer.

## H  RAW RESULTS OF EXPERIMENTS

For both presentational and page layout reasons we have chosen to present most of results in the form of charts in the body or the article. Considering that exact results are important for both reproducibility as well as future comparisons to our work, we provide below the raw results of our experiments in the form of detailed tables. For completeness, some of the tables we did include in the body of the article are duplicated to here as well.

|  | $N = 0$ | 25 | 50 | 75 | 100 | 125 | 150 |
|---|---|---|---|---|---|---|---|
| LP-Based | 97.9 | 97.5 | 96.4 | 94.1 | 89.2 | 80.9 | 70.2 |
| GHT-model | **98.5** | **98.2** | **97.8** | **96.5** | **93.9** | **87.1** | **76.3** |

Table 3: Blind classification with missing data on the binary MNIST dataset with feature deletion noise according to Globerson and Roweis (2006), averaged over all pairs of digits.

|  | $p = 0$ | 0.25 | 0.50 | 0.75 | 0.90 | 0.95 | 0.99 |
|---|---|---|---|---|---|---|---|
| KNN | 96.8 | 96.7 | 96.2 | 94.4 | 86.4 | 71.7 | 29.2 |
| Zero † | **99.2** | 97.3 | 88.2 | 58.6 | 28.7 | 19.5 | 12.6 |
| Mean † | **99.2** | 98.4 | 90.9 | 52.4 | 21.1 | 15.6 | 10.9 |
| GSN † | **99.2** | 97.4 | 88.5 | 51.8 | 17.7 | 12.6 | 10.1 |
| NICE † | **99.2** | 98.9 | 97.9 | 82.6 | 36.3 | 20.2 | 11.7 |
| DPM † | **99.2** | **99.0** | 98.2 | 89.4 | 47.7 | 25.7 | 12.7 |
| MP-DBM * | 99.0 | 98.0 | 97.0 | 92.0 | 35.0 | 18.0 | 13.0 |
| GCP-model | 96.6 | 96.4 | 95.7 | 92.2 | 79.8 | 66.5 | 31.2 |
| GHT-model | 99.0 | **99.0** | **98.7** | **97.7** | **90.5** | **76.0** | **33.0** |

Table 4: Blind classification with missing data on the multi-class MNIST dataset, generated according to i.i.d. corruption with probability $p$ for each pixel. (*) Accuracies are estimated from the plot presented in Goodfellow et al. (2013). (†) Data imputation algorithms followed by a standard ConvNet.

| $(N, W) =$ | (1,7) | (2,7) | (3,7) | (1,11) | (2,11) | (3,11) | (1,15) | (2,15) | (3,15) |
|---|---|---|---|---|---|---|---|---|---|
| KNN | 96.6 | 94.0 | 87.1 | 95.9 | 90.3 | 76.7 | 95.0 | 86.1 | 65.0 |
| Zero † | 93.0 | 74.9 | 47.6 | 86.2 | 56.2 | 31.2 | 78.6 | 44.2 | 22.6 |
| Mean † | 97.9 | 89.9 | 67.8 | 95.8 | 74.1 | 42.0 | 91.8 | 60.0 | 27.4 |
| GSN † | 97.4 | 86.8 | 56.8 | 94.2 | 64.3 | 31.8 | 88.9 | 46.4 | 21.8 |
| NICE † | 98.5 | 93.2 | 74.9 | 97.7 | 81.3 | 52.3 | 95.7 | 69.1 | 38.0 |
| DPM † | 97.2 | 87.0 | 64.0 | 94.4 | 73.2 | 44.6 | 91.4 | 61.8 | 33.2 |
| GCP-model | 96.0 | 93.1 | 85.0 | 95.1 | 88.7 | 73.3 | 94.5 | 83.7 | 62.4 |
| GHT-model | **98.6** | **97.3** | **91.2** | **98.3** | **93.7** | **79.1** | **98.0** | **89.6** | **67.2** |

Table 5: Blind classification with missing data on the multi-class MNIST dataset, generated according to missing rectangles corruption with $N$ missing rectangles, each of width and hight equal to $W$. († ) Data imputation algorithms followed by a standard ConvNet.

| | $p = 0$ | 0.25 | 0.50 | 0.75 | 0.90 | 0.95 | 0.99 |
|---|---|---|---|---|---|---|---|
| KNN | 81.3 | 81.0 | 80.8 | 80.4 | **78.0** | **74.4** | **55.6** |
| Zero † | **96.8** | 19.3 | 19.7 | 20.0 | 20.0 | 20.0 | 19.7 |
| Mean † | **96.8** | 66.8 | 49.7 | 35.5 | 30.2 | 24.2 | 20.1 |
| NICE † | **96.8** | 95.8 | 91.5 | 70.7 | 30.9 | 22.9 | 20.5 |
| DPM † | **96.8** | 88.8 | 60.2 | 28.2 | 21.3 | 20.9 | 20.6 |
| GHT-model | 96.7 | **96.6** | **94.9** | **84.0** | 67.9 | 58.1 | 41.2 |

Table 6: Blind classification with missing data on the multi-class NORB dataset, generated according to i.i.d. corruption with probability $p$ for each pixel. († ) Data imputation algorithms followed by a standard ConvNet.

| $(N, W) =$ | (1,7) | (2,7) | (3,7) | (1,11) | (2,11) | (3,11) | (1,15) | (2,15) | (3,15) |
|---|---|---|---|---|---|---|---|---|---|
| KNN | 81.2 | 81.0 | 81.0 | 81.1 | 80.4 | 79.8 | 80.5 | 78.4 | 75.3 |
| Zero † | 35.9 | 28.1 | 25.1 | 25.7 | 22.6 | 20.9 | 22.4 | 20.5 | 19.8 |
| Mean † | 81.9 | 73.0 | 66.6 | 63.2 | 49.6 | 41.9 | 45.7 | 32.5 | 25.9 |
| NICE † | 96.1 | 95.3 | 93.7 | 92.1 | 81.4 | 67.4 | 73.8 | 46.4 | 33.0 |
| DPM † | 90.1 | 81.9 | 74.2 | 65.9 | 46.0 | 34.3 | 37.7 | 24.2 | 20.9 |
| GHT-model | **96.5** | **96.3** | **95.9** | **95.5** | **93.7** | **91.2** | **92.3** | **86.0** | **79.4** |

Table 7: Blind classification with missing data on the multi-class NORB dataset, generated according to missing rectangles corruption with $N$ missing rectangles, each of width and hight equal to $W$. († ) Data imputation algorithms followed by a standard ConvNet.

| $p_{\text{train}}$ \ $p_{\text{test}}$ | 0.25 | 0.50 | 0.75 | 0.90 | 0.95 | 0.99 |
|---|---|---|---|---|---|---|
| 0.25 | 98.9 | 97.8 | 78.9 | 32.4 | 17.6 | 11.0 |
| 0.50 | **99.1** | 98.6 | 94.6 | 68.1 | 37.9 | 12.9 |
| 0.75 | 98.9 | **98.7** | **97.2** | 83.9 | 56.4 | 16.7 |
| 0.90 | 97.6 | 97.5 | 96.7 | **89.0** | 71.0 | 21.3 |
| 0.95 | 95.7 | 95.6 | 94.8 | 88.3 | **74.0** | 30.5 |
| 0.99 | 87.3 | 86.7 | 85.0 | 78.2 | 66.2 | **31.3** |
| i.i.d. (rand) | 98.7 | 98.4 | 97.0 | 87.6 | 70.6 | 29.6 |
| rects (rand) | 98.2 | 95.7 | 83.2 | 54.7 | 35.8 | 17.5 |

Table 8: We compare ConvNets on the MNIST dataset, trained on i.i.d. corruption with probability $p_{\text{train}}$ while tested on i.i.d. corruption with probability $p_{\text{test}}$. Additionally, we trained ConvNets on either i.i.d. or missing rectangles corruption distributions with random corruption parameters sampled for each batch of training samples, while testing on i.i.d. corruption with the fixed parameter $p_{\text{test}}$.

| test<br>train | (N,W)=(1,8) | (1,12) | (1,16) | (2,8) | (2,12) | (2,16) | (3,8) | (3,12) | (3,16) |
|---|---|---|---|---|---|---|---|---|---|
| rects (fixed) | 98.7 | **97.7** | **93.1** | **98.6** | **94.7** | **82.0** | **98.2** | **90.5** | **70.5** |
| rects (rand) | **99.0** | 97.6 | 92.3 | 98.4 | 94.6 | 80.1 | 98.0 | 90.0 | 66.9 |
| i.i.d. (rand) | 97.8 | 94.8 | 83.4 | 96.8 | 88.6 | 64.5 | 96.1 | 80.6 | 49.5 |

Table 9: We compare ConvNets on the MNIST dataset, train and tested on the same (fixed) missing rectangles distribution, against ConvNets trained on randomly chosen missingness distributions from either the missing rectangles or i.i.d. corruption distributions.

