# Peer review of "Tensorial Mixture Models"

_ICLR 2017 — rejected_

[Official Review · AnonReviewer4 · rating 5 · confidence 3 · 15 Dec 2016]
**The ideas are brilliant, but technical typos exist**

The paper provides an interesting use of generative models to address the classification with missing data problem. The tensorial mixture models proposed take into account the general problem of dependent samples. This is an nice extension of current mixture models where samples are usually considered as independent. Indeed the TMM model is reduced to the conventional latent variable models. As much as I love the ideas behind the paper, I feel pitiful about the sloppiness of the presentation (such as missing notations) and flaws in the technical derivations.  Before going into the technical details, my high level concerns are as follows:
(1) The joint density over all samples is modeled as a tensorial mixture generative model. The interpretation of the CP decomposition or HT decomposition on the prior density tensor is not clear. The authors have an interpretation of TMM as product of mixture models when samples are independent, however their interpretation seems flawed to me, and I will elaborate on this in the detailed technical comments below. 
(2) The authors employ convolution operators to compute an inner product. It is realizable by zero padding, but the invariance structure, which is the advantage of CNN compared to feed-forward neural network, will be lost. However, I am not sure how much this would affect the performance in practice. 
(3) The author could comment in the paper a little bit on the sample complexity of this method given the complexity of the model. 

Because I liked the ideas of the paper so much, and the ICLR paper submitted didn't present the technical details well due to sloppiness of notations, so I read the technical details in the arXiv version the authors pointed out. There are a few technical typos that I would like to point out (my reference to equations are to the ones in the arXiv paper). 
(1) The generative model as in figure (5) is flawed. P(x_i|d_i;\theta_{d_i}) are vectors of length s, there the product of vectors is not well defined. It is obvious that the dimensions of the terms between two sides of the equation are not equal. In fact, this should be a tucker decomposition instead of multiplication. It should be P(X) = \sum_{d1,\ldots,d_N} P(d_1,\ldots,d_N) (P(x_1|d_1;theta_{d_1},P(x_2|d_2;theta_{d_2},\ldots,P(x_N|d_N;theta_{d_N}), which means a sum of multi-linear operation on tensor P(d_1,\ldots,d_N), and each mode is projected onto P(x_i|d_i;theta_{d_i}. 
(2) I suspect the special case for diagonal Gaussian Mixture Models has some typos as I couldn't derive the third last equation on page 6. But it might be just I didn't understand this example. 
(3) The claim that TMM reduces to product of mixture model is not accurate. The first equation on page 7 is only right when "sum of product" operation is equal to "product of sum" operation. Similarly, in equation (6), the second equality doesn't hold unless in some special cases. However, this is not true. This might be just a typo, but it is good if the authors could fix this. I also suspect that if the authors correct this typo,the performance on MNIST might be improved.

Overall, I like the ideas behind this paper very much. I suggest the authors fix the technical typos if the paper is accepted.

[Official Review · AnonReviewer1 · rating 7 · confidence 3 · 16 Dec 2016]
**Interesting approach to generative models and missing data**

This paper uses Tensors to build generative models.  The main idea is to divide the input into regions represented with mixture models, and represent the joint distribution of the mixture components with a tensor.  Then, by restricting themselves to tensors that have an efficient decomposition, they train convolutional arithmetic circuits to generate the probability of the input and class label, providing a generative model of the input and labels.  

This approach seems quite elegant.  It is not completely clear to me how the authors choose the specific architecture for their model, and how these choices relate to the class of joint distributions that they can represent, but even if these choices are somewhat heuristic, the overall framework provides a nice way of controlling the generality of the distributions that are represented.

The experiments are on simple, synthetic examples of missing data.  This is somewhat of a limitation, and the paper would be more convincing if it could include experiments on a real-world problem that contained missing data.  One issue here is that it must be known which elements of the input are missing, which somewhat limits applicability.  Could experiments be run on problems relating to the Netflix challenge, which is the classic example of a prediction problem with missing data?  In spite of these limitations, the experiments provide appropriate comparisons to prior work, and form a reasonable initial evaluation.

I was a little confused about how the input of missing data is handled experimentally.  From the introductory discussion my impression was that the generative model was built over region patches in the image.  This led me to believe that they would marginalize over missing regions.  However, when the missing data consists of IID randomly missing pixels, it seems that every region will be missing some information.  Why is it appropriate to marginalize over missing pixels?  Specifically, $x_i$ in Equation 6 represents a local region, and the ensuing discussion shows how to marginalize over missing regions.  How is this done when only a subset of a region is missing?  It also seems like the summation in the equation following Equation 6 could be quite large.  What is the run time of this? 

The paper is also a bit schizophrenic about the extent to which the results are applicable beyond images.  The motivation for the probabilistic model is mostly in terms of images.  But in the experiments, the authors state that they do not use state-of-the-art inpainting algorithms because their method is not limited to images and they want to compare to methods that are restricted to images.  This would be more convincing if there were experiments outside the image domain.

It was also not clear to me how, if at all, the proposed network makes use of translation invariance.  It is widely assumed that much of the success of CNNs comes from their encoding of translation invariance through weight sharing.   Is such invariance built into the authors’ network?  If not, why would we expect it to work well in challenging image domains?

As a minor point, the paper is not carefully proofread.  To just give a few examples from the first page or so:

“significantly lesser” -> “significantly less”

“the the”

“provenly” -> provably

[Official Review · AnonReviewer3 · rating 4 · confidence 4 · 21 Dec 2016]
**Interesting approach but ...**

This paper proposes a generative model for mixtures of basic local structures where the dependency between local structures is a tensor. They use tensor decomposition and the result of their earlier paper on expressive power of CNNs along with hierarchical Tucker to provide an inference mechanism. However, this is conditioned on the existence of decomposition. The authors do not discuss how applicable their method is for a general case, what is the subspace where this decomposition exists/is efficient/has low approximation error. Their answer to this question is that in deep learning era these theoretical analysis is not needed. While this claim is subjective, I need to emphasize that the paper does not clarify this claim and does not mention the restrictions. Hence, from theoretical perspective, the paper has flaws and the claims are not justified completely. Some claims cannot be justified with the  current results in tensor literature as the authors also mentioned in the discussions. Therefore, they should have corrected their claims in the paper and made the clarifications that this approach is restricted to a clear subclass of tensors.

If we ignore the theoretical aspect and only consider the paper from empirical perspective, the experiments the appear in the paper are not enough to accept the paper. MNIST and CIFAR-10 are very simple baselines and more extensive experiments are required. Also, the experiments for missing data are not covering real cases and are too synthetic. Also, the paper lacks the extension beyond images. Since the authors repeatedly mention that their approach goes beyond images, and since the theory part is not complete, those experiments are essential for acceptance of this paper.

[Final Decision · Program Chairs · 06 Feb 2017]
**ICLR committee final decision**

The authors have recently made several connections between deep learning and tensor algebra. While their earlier works dealt with supervised learning, the current work analyzes generative models through the lens of tensor algebra. 
 The authors show propose a tensorial mixture model over local structures where the mixture components are expressed as tensor decompositions. They show that hierarchical tensor decomposition is exponentially more expressive compared to the shallow models. 
 The paper makes original contributions in terms of establishing expressivity of deep generative models. The connections with tensor algebra could lead to further innovations, e.g. in training algorithms.
 However, the paper can be improved in two aspects: 
 (1) It will be nice if the authors make a connection between the algebraic view presented here with the geometric view presented by: